# Kinematic gait characteristics of straight line walk in clinically sound dairy cows

M. Tijssen[1]*, F. M. Serra Bragança[2], K. Ask[3], M. Rhodin[3], P. H. Andersen[3], E. Telezhenko[4], C. Bergsten[5], M. Nielen[1], E. Hernlund[3]

1 Department of Population Health Sciences, Faculty of Veterinary Medicine, Utrecht University, Utrecht, The Netherlands, 2 Department of Clinical Sciences, Faculty of Veterinary Medicine, Utrecht University, Utrecht, The Netherlands, 3 Department of Anatomy, Physiology and Biochemistry, Swedish University of Agricultural Sciences, Uppsala, Sweden, 4 Department of Biosystems and Technology, Swedish University of Agricultural Sciences, Alnarp, Sweden, 5 Department of Clinical Sciences, Swedish University of Agricultural Sciences, Uppsala, Sweden

* m.tijssen@uu.nl

**Data Availability Statement:** All relevant data are within the paper and its S1 Table and S1 File and S1–S8 Figs.

## Abstract

The aim of this study is to describe the kinematic gait characteristics of straight line walk in clinically sound dairy cows using body mounted Inertial Measurement Units (IMUs) at multiple anatomical locations. The temporal parameters used are speed and non-speed normalized stance duration, bipedal and tripedal support durations, maximal protraction and retraction angles of the distal limbs and vertical displacement curves of the upper body. Gait analysis was performed by letting 17 dairy cows walk in a straight line at their own chosen pace while equipped with IMU sensors on tubera sacrale, left and right tuber coxae (LTC and RTC), back, withers, head, neck and all four lower limbs. Data intervals with stride by stride regularity were selected based on video data. For temporal parameters, the median was calculated and 95% confidence intervals (CI) were estimated based on linear mixed model (LMM) analysis, while for limb and vertical displacement curves, the median and most typical curves were calculated. The temporal parameters and distal limb angles showed consistent results with low variance and LMM analysis showed non-overlapping CI for all temporal parameters. The distal limb angle curves showed a larger and steeper retraction angle range for the distal front limbs compared with the hind limbs. The vertical displacement curves of the sacrum, withers, LTC and RTC showed a consistent sinusoidal pattern while the head, back and collar curves were less consistent and showed more variation between and within cows. This kinematic description might allow to objectively differentiate between normal and lame gait in the future and determine the best anatomical location for sensor attachment for lameness detection purposes.

## Introduction

The kinematic study of gait characteristics in dairy cows can provide important insight into the normal walking gait patterns, allowing us to objectively differentiate normal from abnormal gait [1, 2]. Lameness can be defined as a deviation from the normal gait pattern due to

**Funding:** The study was funded by the grants 2016-01760 (MR) and 2018-00737 (EH) from the Swedish Research Council Formas, (http://www.formas.se/). The funders had no role in study design, data collection and analysis, decision to publish, or preparation of the manuscript. Indirect support was provided through salaries by the home institutions of all co-authors. MT received an Erasmus+ grant for her travel to visit the Swedish University of Agricultural Sciences, Uppsala, Sweden.

**Competing interests:** The authors have declared that no competing interests exist.

compensatory movements to avoid any pain or discomfort [3]. Despite intense research, lameness still is one of the most important welfare issues in dairy cows since its prevalence is high and it often remains undetected until the lameness degree worsens [4–6]. Therefore, recognition and treatment of lameness at an early stage improves welfare for the cow and also reduces the economical and unwanted consequences for the farmer [7, 8].

Currently, lameness detection is based on visual assessment using subjective locomotion scoring to recognize alterations in the gait patterns [9–12]. These locomotion scales mainly make use of stride change, body poses and abnormal position of the back. However, studies show that farmers visually recognize only a third of the lame cows in their herd, which are assessed as severely lame cows by researchers [12, 13]. A method with high sensitivity for subtle lameness is therefore warranted [2, 14]. Research has been focused on refinement of behavioral methods, based on quantification of lying and standing behaviors, which also proved insensitive for early detection of subtle lameness [15, 16]. Several works are describing the kinematic gait characteristics of both lame and clinically sound cows [17–19]. And, only recently, gait characteristics were introduced in combination with the behavioral-based methods [2, 14], revealing a need for more knowledge on kinematic gait characteristics in cows to allow objectively differentiation of the normal gait from abnormal gait.

In equine practice, quantitative lameness detection has matured into a practical clinical tool, and is routinely performed using optical motion systems or Inertial Measurement Units (IMUs) [3]. These techniques have overcome the low accuracy inherent to the human visual assessment by measuring subtle changes in kinematic gait characteristics [20–22]. The most reliable and clinically used characteristics for lameness assessment in trot is the vertical excursion of the head, withers and tubera sacrale measured by IMUs attached to the upper body [20, 23, 24]. This vertical excursion is also useful for detection of lameness at the walk, together with temporal stride parameters (e.g. stride and stance duration) and joint angles of the limbs [23, 25–27].

Since many cows already wear single low-resolution accelerometers around their neck or limb, it is most obvious to focus on accelerometer based techniques, such as IMUs, to overcome major barriers for adoption on dairy farms [2]. In a previous study, the gait characteristics of walking dairy cows were based on data from two accelerometers attached to the lateral claw and metatarsus [1]. This study showed promising results and a description of gait characteristics that differ between lame and non-lame cows [1]. However, vertical excursion of the upper body has not yet been described of the walking gait in dairy cows. Given the visibility of these body landmarks, even for a cow among herd mates, it would be of future interest to explore if asymmetries in the vertical excursions of the upper body could prove sensitive for detection of lame animals. This can be of particular interest in the development on automated lameness detection systems based on computer vision, where challenging sensor attachment is superfluous, but optical occlusion of limbs is a challenge. Prerequisite for such studies is the knowledge of normal walking gait.

The aim of this study is to describe the kinematic gait characteristics and their normal variation of straight line walk in clinically sound dairy cows using body mounted IMUs at multiple anatomical locations. This detailed kinematic description might allow us to objectively differentiate between normal and lame gait characteristics in the future and determine the best anatomical location for sensor attachment for lameness detection purposes.

## Materials and methods

### Ethical statement

The study was approved by the Swedish Ethics Committee and according to the Swedish legislation on animal experiments (diary number 5.8.18-10570/2019).

## Study protocol

Gait analysis was performed during straight line locomotion by letting cows walk on a 72 meter concrete corridor in their own chosen pace while equipped with IMU sensors on different anatomical landmarks. Intervals with stride by stride regularity were selected from the data based on video data for further analysis. Subsequently, temporal parameters as well as distal limb angles and vertical displacement curves were extracted from respectively the limb and upper body IMUs. The temporal parameters used are the stance duration, speed normalized stance duration, bipedal and tripedal support durations, speed normalized bipedal and tripedal support durations, maximal protraction and retraction angles of the distal limbs.

## Experimental animals

For the study, 17 early or mid-lactation cows were selected from the Swedish Livestock Research Centre Lövsta at the Swedish University of Agricultural Sciences (nine Swedish Red and eight Swedish Holstein cows), for more details see S1 Table. The cows were selected if they met the following inclusion criteria: i. they were claw trimmed within the last three months prior to the measurements during which no clinically significant claw disorders were recorded, ii. they showed no signs of pain, nervous or stressed behavior, iii. they were assessed as clinically healthy and scored zero on the Sprecher lameness scale [9] as evaluated by two experienced raters (CB and ET), iv. they had a body exterior within normal range and a normal limb exterior without major visible deviations.

## IMUs and sensor placement

The measurements were performed with the equine gait analysis system EquiMoves® [28]. The cows were equipped with 11 ProMove-mini wireless IMU sensors (Inertia-Technology B.V., Enschede) as can be seen in Fig 1. The sensors were placed on the following anatomical landmarks; just caudal to the nuchal crest (further called head), the highest point of the withers (further called withers), the spinal process of the 13th thoracic vertebra (further called back), between the tubera sacrale of the pelvis (further called sacrum), right and left tuber coxae (further called RTC and LTC respectively), lateral aspect of the mid metatarsus/metacarpus of each limb (further called limb or LF (left front), RF (right front), LH (left hind), RH (right hind)) and one sensor was attached to the inner right side of neck collar (further called collar). The upper body sensors were attached to the skin with cyanoacrylate glue, and limb and collar sensors were attached with straps. After the measurements, the sensors were removed with acetone from the upper body.

The IMU sensors attached to the upper body were set to a range of ± 8 *g* for the low-*g* acceleration, ± 100 *g* for the high-*g* acceleration and 2000 degrees/s for the angular velocity. The IMU sensors attached to the limbs were set to a range of 16 *g* for the low-*g* acceleration, 200 *g* for the high-*g* acceleration and 2000 degrees/s for the angular velocity. All sensors were set to a sampling rate of 200 Hz and synchronized in time with an accuracy of < 100 ns.

The IMU sensors were calibrated when every measurement was started with a period of five seconds of silent signal in which the cow was completely standing still. After each measurement, the acceleration and angular velocity data (further described as IMU data) were wirelessly transmitted via the Inertia Gateway to the Inertia Studio software (version 3.5.2).

## Data collection

At each day of the measurements, two cows were moved to the stable where the measurements took place to familiarize themselves with the surroundings. The cows were allowed to walk freely and had access to hay and water ad libitum.

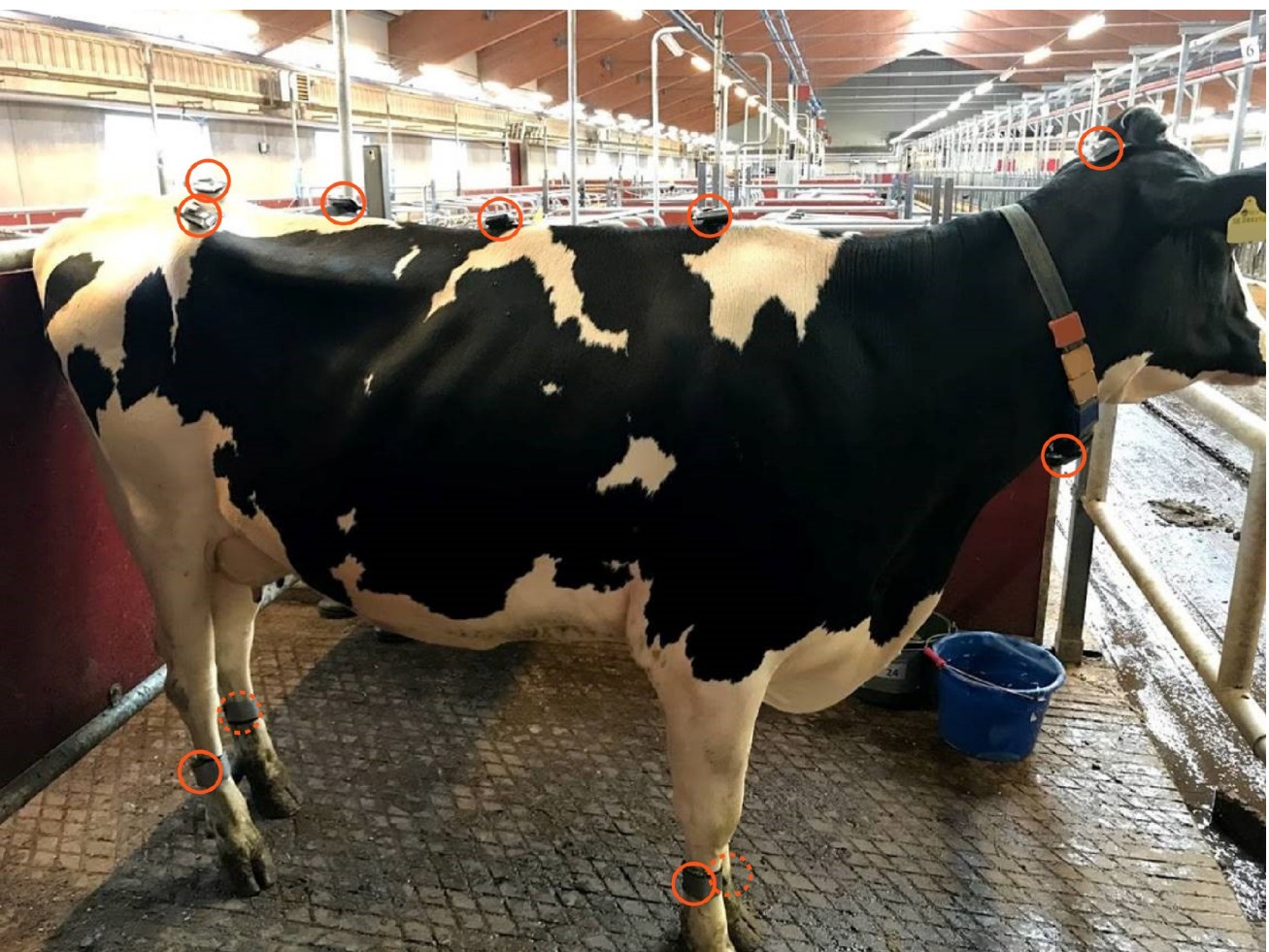

**Fig 1. Cow equipped with IMU sensors on predefined anatomical landmarks.** IMUs are indicated with orange circles; just caudal to the nuchal crest, the highest point of the withers, the spinal process of the 13th thoracic vertebra, between the tubera sacrale of the pelvis, right and left tuber coxae, lateral aspect of the metatarsus/metacarpus of each limb and one sensor was attached on the inner right side of neck collar.

For the measurements, each cow was walked through a corridor of 72 meters long with diamond grooved solid concrete flooring and cubicles and fences on both sides. The cows had to walk this corridor up and down twice (in total 288 meters) with a self-chosen calm and constant pace to obtain data of a natural walking flow. One or two researchers walked behind the cow from a distance to prevent the cow from turning around, while handling the cow as little as possible to not disturb the measurements. No other animals were present in the corridor during measurements.

Each measurement was video recorded from the side by a researcher walking alongside the cow with a handheld camera (Canon Legria HF R78, Canon, Tokyo, Japan). The handheld camera had a frame rate of 50 fps and a resolution of 1280x720 pixels. The camera was synchronized with the IMUs by tapping the withers sensor twice at the start of the measurement while filming.

## Data selection

From each measurement, selection of IMU data for further analysis was based on video images of the handheld camera and footfall figures (Fig 2; adapted from horses [29]). Intervals were

**Fig 2. Footfall figure of selected interval of a measurement.** Limbs are indicated by separate colors (green: LF, blue: RF, purple: LH, red: RH); colored areas indicate stances and white areas indicate swing phases as adapted from horses [29]. The first box shows an excluded epoch in which three limbs have a prolonged stance duration compared with the LH limb, the second box indicates an included epoch of stride-by-stride regularity in which all limbs have comparable stance and swing durations and the third box indicates an excluded epoch in which three limbs have longer stance durations but not at the same moment in time.

selected when they met the following criteria: i. the cows were walking in a straight line up and down the corridor (no turns), ii. the cows were physical handled to walk by the driver (to avoid selection of intervals with anomalous walking patterns or body postures), iii. stride by stride regularity was seen in the footfall figures (the second box of Fig 2). A maximum of four intervals per measurement was used for analysis.

## Data analysis

The selected intervals from the IMU data were exported to MATLAB (version R2017a, The MathWorks Inc., Natick, Massachusetts, USA) for analysis by the following steps which are also shown in S1 Fig.

Claw-on and claw-off timings were detected with already existing algorithms developed for horses [30] and these timings were used for stride segmentation of the limb data. Stride segmentation was performed based on the claw-on moment of the LH limb as previously described [28].

For the limb data, the sagittal orientation of the IMU was calculated as described in the EquiMoves data analysis framework [28] and based on the quaternion-based complementary filter [31]. This resulted in stride segmented distal limb angle curves from which the maximal distal limb angles were calculated. Maximal protraction is the maximal forward extension (positive angle) from midstance and maximal retraction is the maximal backward extension (negative angle) from midstance of a distal limb in the sagittal plane (S2 Fig; adapted from horses [32]). These distal limb angles and stride segmented limb angle curves were used for distal limb angle analysis.

For the not stride segmented upper body IMU signals, the orientation on a global coordinate frame was calculated based on the quaternion-based complementary filter [31]. Thereafter, the vertical displacement signals were determined based on a cyclic double integration of the acceleration signal [33] and used for the vertical displacement analysis. Stride segmentation of these signals was performed with a novel method as described in the vertical displacement analysis section.

**Temporal analysis.** From the stride segmented limb data, the stance duration was calculated for all limbs separately while the bipedal and tripedal support durations were calculated for all possible limb (pairs). The definitions of these durations can be found Table 1. Since the cows were walking in their own chosen pace, speed normalization was performed for all these durations by dividing the duration with the entire stride duration of the corresponding stride of the LH limb resulting in a fraction (i.e. duty factor). All the calculated durations and fractions were evaluated and durations shorter than zero seconds and fractions bigger than one or smaller than zero were excluded from further analysis.

Distributions of these parameters were visualized in boxplots for every cow and limb (pairs) separately. The median duration of each parameter and their 95% confidence interval

**Table 1. Detailed description of the calculated parameters used in this study.** Limbs are indicated by LF (left front), RF (right front), LH (left hind) and RH (right hind). These definitions are adapted from horses [25].

| Name | Abbreviation | Description |
|---|---|---|
| Stance duration (sec) | | Time between claw-on and subsequent claw-off |
| Stride duration (sec) | | Time between claw-on and subsequent claw-on |
| Bipedal support phase duration | | Duration of simultaneous stance phase of two limbs: |
| (sec) | LF-RH | left diagonal |
| | RF-LH | right diagonal |
| | LF-LH | left ipsilateral |
| | RF-RH | right ipsilateral |
| Tripedal support phase duration | | Duration of simultaneous stance phase of three limbs: |
| (sec) | not LF | RF-LH-RH |
| | not RF | LF-LH-RH |
| | not LH | LF-RF-RH |
| | not RH | LF-RF-LH |
| Speed normalization | | Duration divided by entire stride duration of LH (resulting in a fraction between 0 and 1). |
| Distal limb angles | max protraction | Maximal forward protraction of the distal limb measured at the metacarpus/-tarsus in the sagittal plane |
| (degrees) | max retraction | Maximal backward retraction of the distal limb measured at the metacarpus/-tarsus in the sagittal plane |
| Vertical displacement (mm) | | Upwards and downwards displacement of the sensor in the sagittal plane |

was calculated by bootstrapping. Differences between limbs, for stance duration, and limb pair combinations, for bipedal and tripedal support durations, were evaluated based on the stride level data by linear mixed model analysis with the "lme4" package [34] in R (version 1.1.442, RStudio Inc., Boston, Massachusetts, USA). For these models, limb (pairs) and deviations from the median stride duration was taken as fixed effects, except for the speed normalized data, and cow as random effect. Deviations of normality and homoscedasticity of the residuals was visually checked by examining the QQ plot and residual plot. Thereafter, 95% confidence intervals were evaluated with the "confint" package [35] and used for hypothesis testing.

**Distal limb angle analysis.** The determined distal limb angles were evaluated; maximal retraction angles larger than zero degrees and maximal protraction angles smaller than zero degrees were excluded. Visualization and estimation of the confidence intervals was performed in the same manner as described for the temporal parameters.

The stride segmented distal limb angle curves were linear interpolated to 100 samples to ensure that all curves were of the same length, allowing comparison between and within cows since the cows were walking in their own chosen pace. The following analysis steps were performed over all cows and steps for every limb separately: i) the differences between all the curves was calculated and the curve with the least difference is further called the most typical curve, ii) the median curve and median absolute deviation (MAD), mean claw-on timing and mean claw-off timing were calculated, and iii) the curves were depicted on a scale from zero to 100; where zero indicated the start of the stride (first claw-on moment) and 100 the end of the stride (next claw-on moment).

**Vertical displacement analysis.** The vertical displacement signals of the upper body sensors (sacrum, RTC, LTC, back, withers, head and collar) were cut into intervals from claw-on to claw-on timing of the left hind limb. Before and after these timings, ten percent of the entire

stride duration was added, to make visual inspection of the intervals more intuitive, and every interval was linear interpolated to 100 samples allowing comparison between and within cows. The following analysis steps were performed over all steps for every cow and sensor location separately: i) the most typical curve, median curve and MAD were calculated, ii) the mean claw-on and claw-off timings were calculated for every limb, to indicate the stance and swing phase for every limb in the figures (further called footfalls), and iii) the curves and footfalls were depicted on a scale from zero to 100; zero indicates the start of the stride (first claw-on moment of the left hind limb) and 100 indicates the end of the stride (second claw-on moment of the left hind limb).

## Results and discussion

### Data description

In total, 32 measurements were performed in 17 cows, with on average two measurements per cow. The characteristics of the cows are displayed in S1 Table. The number of selected strides per cow ranged between 72 and 286 with a median of 204.

For the temporal analysis, a total of two stance durations, one stride duration and 63 normalized stance durations were excluded. The high number of excluded normalized stance durations was caused by an unequal number of strides per limb within one interval, as can be seen in Fig 2. The temporal data were not normally distributed on a cow level. No data was excluded during the distal limb angle and vertical displacement analysis.

### Temporal and distal limb angle analysis

The distribution of the stance duration, in seconds, is shown on the left side and the speed normalized stance duration, as fraction of the entire stride duration, is shown on the right side of Fig 3.

Fig 3 suggests similar and consistent stance durations between 0.5 and 1 s and speed normalized stance duration between 0.50 and 0.75 for all cows. For the stance duration, a median value of 0.87 s was found over all cows and limbs, with a longer median around 0.91 s for the front limbs and 0.86 s for the hind limbs (Table 2). On a cow level, the boxes overlap, and the width of the boxes seem to be rather similar for most cows and their limbs, which indicates that the variance in stance duration might be fairly consistent for all cows and limbs. For the speed normalized stance duration, a median value of 0.66 was found over all cows and limbs, with a longer median around 0.68 for the front limbs and 0.64 for the hind limbs (Table 2). On a cow level, the differences between the front and hind limbs seem to become more obvious after speed normalization, while some cows show a large variation with wider boxes.

The differences in stance duration between the front and hind limbs might suggest a difference in protraction and retraction angle of the distal limbs. In a previous study in cows [1] and in horses [25], a duty factor of around 0.63 was found, which is in agreement with the fraction of the entire stride duration found for the hind limbs in this study.

The distribution of the bipedal support durations, in seconds, is shown on the left side and the speed normalized bipedal durations, as fraction of the entire stride duration, is shown on the right side of Fig 4.

For the bipedal support durations (Fig 4), the figure suggests slightly longer support durations for diagonal limb pairs, although this difference does not seem obvious for all cows. On a group level, a median value of 0.13 s was found, with a longer median value 0.14 s for the diagonal limb pairs and 0.10 s for the ipsilateral limb pairs (Table 2). Furthermore, a greater variation in the ipsilateral durations might be seen for most cows, indicated by wider boxes, which might suggest less consistent ipsilateral support durations for most cows. For the speed

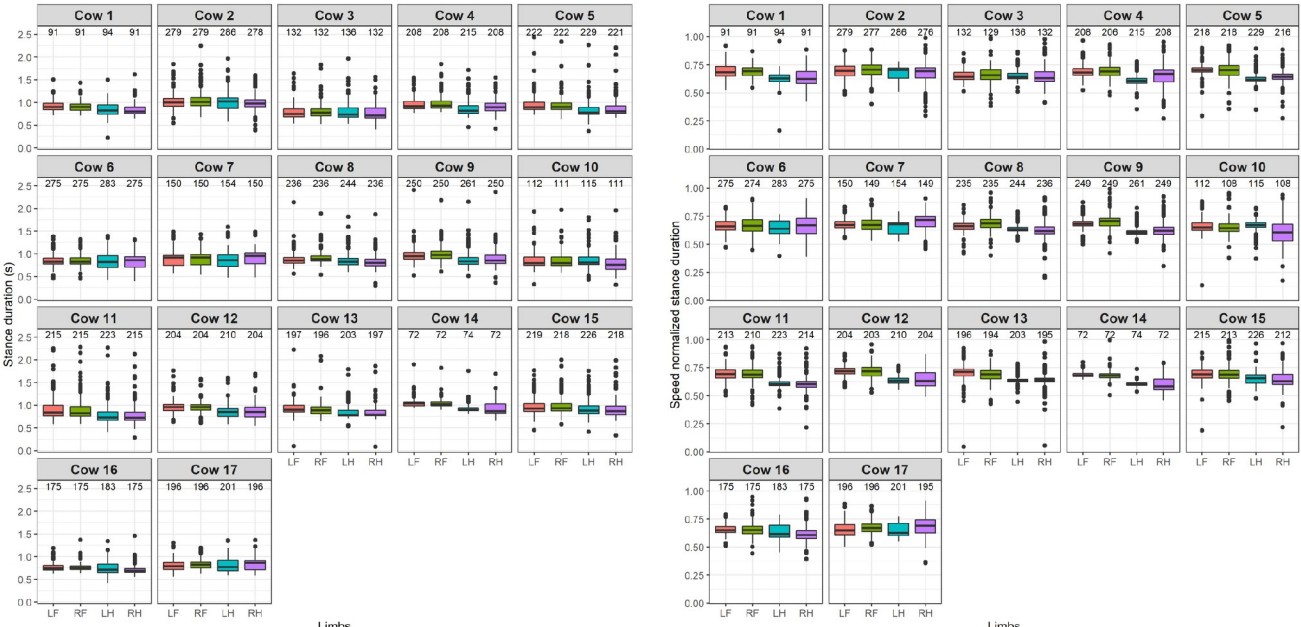

**Fig 3. Distributions of stance durations (left panel) and speed normalized stance durations (right panel) for every cow and limb separately.** In red: LF, green: RF, blue: LH and purple: RH. The number of selected strides per limb is shown above the boxes. The spacing of the boxplot shows the range between the first and third quartile, the median is indicated with the black line, the whiskers show the range of the maxima and minima and the outliers are indicated with dots.

normalized bipedal support durations, the differences between the diagonal and ipsilateral limb pairs seems not to become more obvious and the variation in the ipsilateral support durations seems not to become clearer. On a group level, a median fraction of 0.10 was found, with a median around 0.11 for the diagonal limb pairs and 0.8 for the ipsilateral limb pairs.

The difference between the diagonal and ipsilateral support durations are in contrast to observations in horses for which slightly shorter diagonal support durations were found compared with ipsilateral durations (0.12 versus 0.13 respectively) in the walk [25].

The distribution of the tripedal support durations, in seconds, is shown on the left side and the speed normalized tripedal support durations, as fraction of the entire stride duration, is shown on the right side of Fig 5.

For the tripedal support durations (Fig 5), the figure suggests that the durations might be shorter when only a single front limb is involved in most cows, although these durations seem to show the largest variation, indicated by the differences in width of the boxes. On a group level, a median duration of 0.21 s was found, with a median duration around 0.18 s for involvement of a single front limb and 0.24 s for involvement of a single hind limb (Table 2). For the speed normalized tripedal support durations, the same differences in durations and variations seem to be found. On a group level, a median fraction of 0.16 was found, with a median fraction around 0.14 for involvement of a single front limb and 0.18 for involvement of a single hind limb, indicating again that cows tend to stand longer when both front limbs are on the ground. In horses in the walk, all support durations were shorter, around 0.12 of the entire stride duration [25].

In summary, the LMM analysis showed non-overlapping confidence intervals between the front and hind limbs, diagonal and ipsilateral limb pairs and single front limb and single hind limb involvement for all speed normalized and non-speed normalized durations. All previous temporal analyses included walking speed normalization to compare temporal parameters

**Table 2. Summary of the temporal parameters and distal limb angles.** The median values and 95% confidence intervals based on the LMM analysis are estimated over all cows and steps for every limb (pair) separately. The median values are given in seconds for the non-speed normalized conditions, in fractions of the entire stride duration (i.e. duty factor), and in degrees for the distal limb angles.

| Parameter | Non-speed normalized (s) | Speed normalized (duty factor) |
|---|---|---|
| Stance duration | | |
| Overall | 0.87 (0.86–0.87) | 0.66 (0.66–0.66) |
| LF | 0.91 (0.90–0.92) | 0.68 (0.67–0.69) |
| RF | 0.92 (0.90–0.93) | 0.69 (0.68–0.70) |
| LH | 0.86 (0.84–0.87) | 0.64 (0.63–0.65) |
| RH | 0.86 (0.84–0.88) | 0.64 (0.63–0.65) |
| Bipedal support phase duration | | |
| Overall | 0.13 (0.13–0.13) | 0.10 (0.10–0.10) |
| LF-RH | 0.14 (0.13–0.15) | 0.11 (0.10–0.12) |
| RF-LH | 0.14 (0.13–0.15) | 0.11 (0.10–0.12) |
| LF-LH | 0.10 (0.10–0.11) | 0.08 (0.07–0.09) |
| RF-RH | 0.11 (0.10–0.12) | 0.08 (0.08–0.09) |
| Tripedal support phase duration | | |
| Overall | 0.21 (0.21–0.21) | 0.16 (0.16–0.16) |
| no LF | 0.18 (0.18–0.19) | 0.14 (0.13–0.14) |
| no RF | 0.19 (0.17–0.20) | 0.14 (0.13–0.14) |
| no LH | 0.24 (0.23–0.25) | 0.18 (0.17–0.19) |
| no RH | 0.24 (0.23–0.25) | 0.18 (0.17–0.19) |
| Maximal protraction (degrees) | | |
| Overall | 25 (25–25) | |
| LF | 25 (24–26) | |
| RF | 26 (25–27) | |
| LH | 24 (23–25) | |
| RH | 23 (22–24) | |
| Maximal retraction (degrees) | | |
| Overall | -36 (-37 - -36) | |
| LF | -46 (-44 - -48) | |
| RF | -46 (-44 - -48) | |
| LH | -29 (-28 - -30) | |
| RH | -30 (-28 - -31) | |

between cows. As the cows were not walking on a treadmill or with a handler in a constant speed, fluctuation in walking speed occurred, which is known to influence the durations of these temporal parameters [36]. Walking speed normalization improved illustration of stance duration patterns between the front and hind limbs although for the bipedal and tripedal support durations, less effect was observed. Whether walking speed normalization is needed in the future depends on the changes due to lameness and whether we want to compare between cows or between limbs, for the latter no walking speed normalization is needed. The variance was found to be low indicated by the small confidence intervals for all the temporal parameters due to the high number of analyzed steps.

The distributions of the maximal protraction angles (left) and maximal retraction angles (right) is shown in degrees for every cow and limb separately in Fig 6.

Fig 6 suggest rather similar protraction angles in most of the cows with little variance, indicated by the width of the boxes. On a group level, a median protraction angle of 25 degrees is

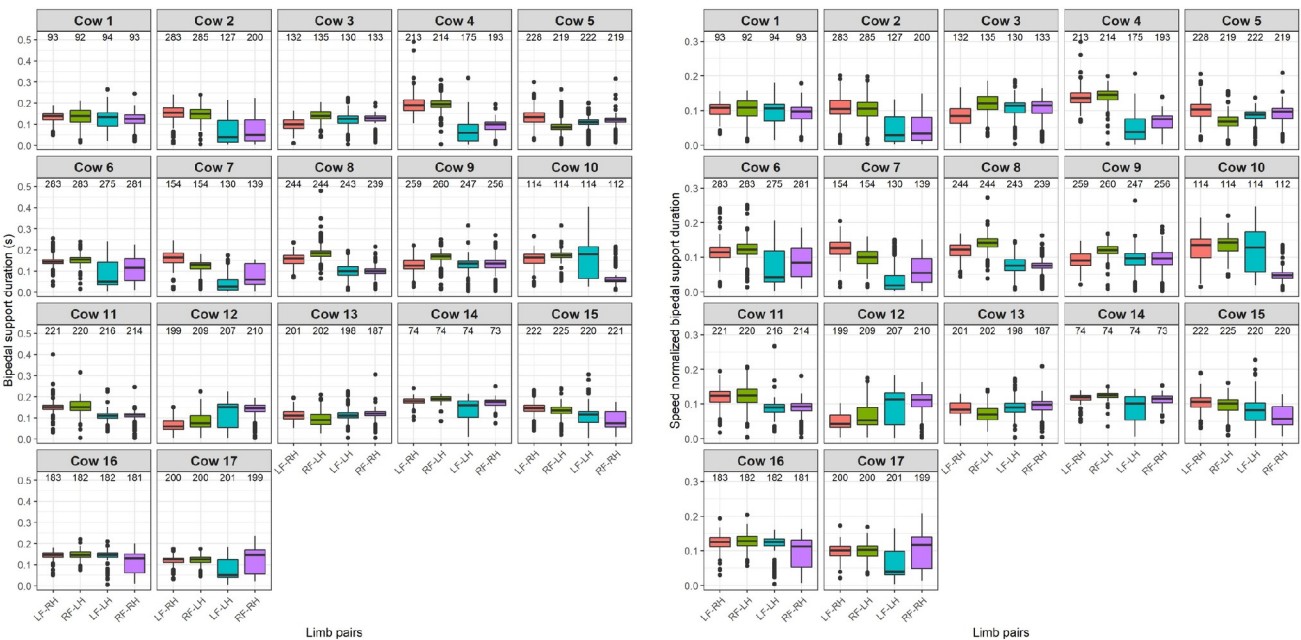

**Fig 4. Distributions of bipedal support durations (left panel) and speed normalized bipedal support durations (right panel) for every cow and limb pairs separately.** In red: LF-RH, green: RF-LH, blue: LF-LH and purple: RF-RH. The number of selected strides per limb is shown above the boxes. The spacing of the boxplot shows the range between the first and third quartile, the median is indicated with the black line, the whiskers show the range of the maxima and minima and the outliers are indicated with dots.

found over all limbs, with a somewhat comparable angle around 25 degrees for the front limbs and around 23 degrees for the hind limbs (Table 2). For the retraction angles, smaller angles

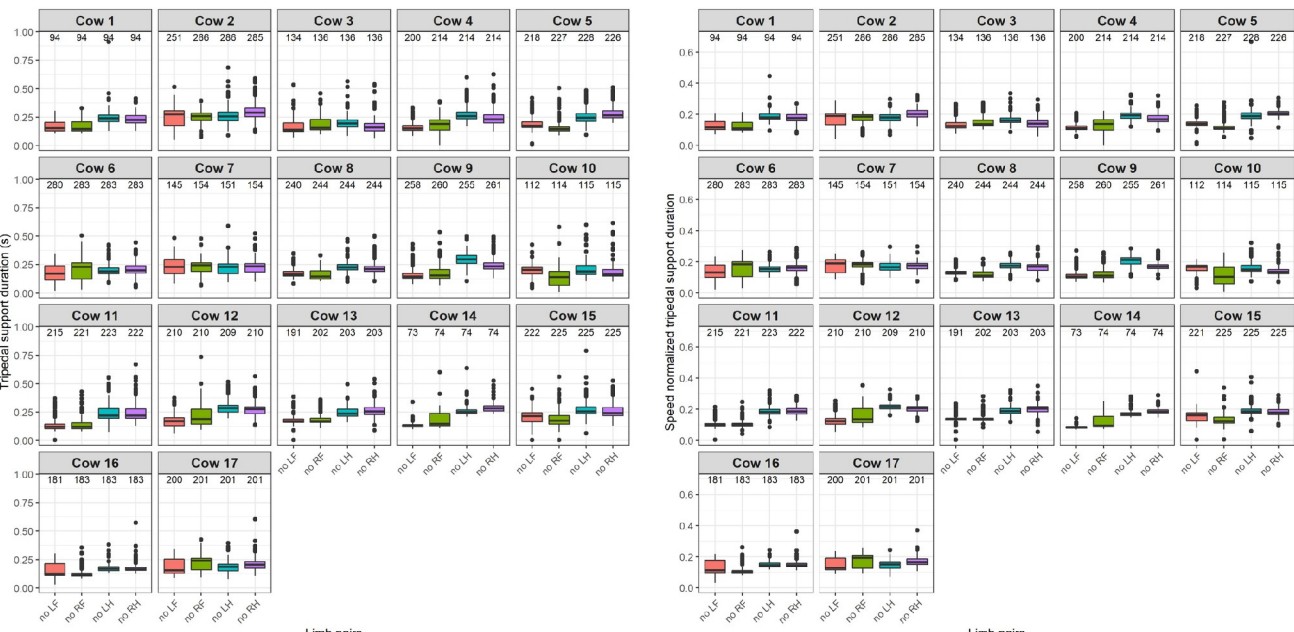

**Fig 5. Distributions of tripedal support durations (left panel) and speed normalized tripedal support durations (right panel) for every cow and limb pair combinations separately.** In red: no LF, green: no RF, blue: no LH and purple: no RH. The number of selected strides per limb is shown above the boxes. The spacing of the boxplot shows the range between the first and third quartile, the median is indicated with the black line, the whiskers show the range of the maxima and minima and the outliers are indicated with dots.

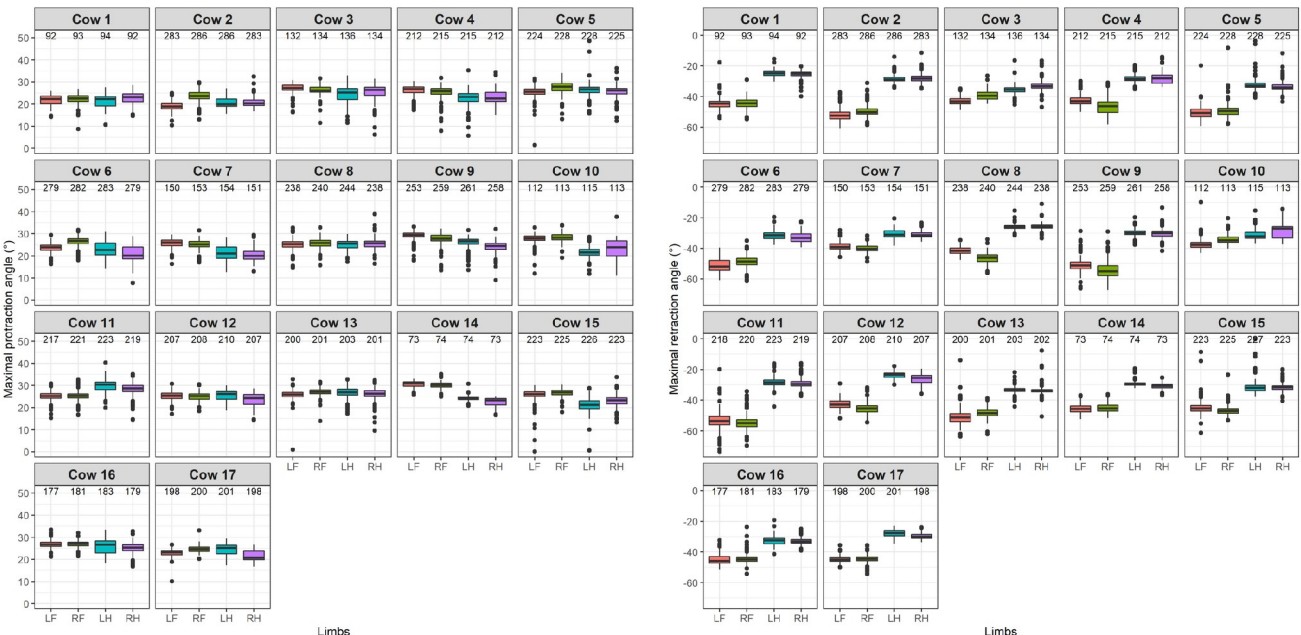

**Fig 6. Distribution of distal limb maximal protraction (left panel) and maximal retraction (right panel) angles in degrees for every cow and limb separately.** In red: LF, green: RF, blue: LH and purple: RH. The number of selected strides per limb is shown above the boxes. The spacing of the boxplot shows the range between the first and third quartile, the median is indicated with the black line, the whiskers show the range of the maxima and minima and the outliers are indicated with dots.

are found for the front limb compared with the hind limbs indicated by the non-overlapping boxes. On a group level, a median retraction angle of -36 degrees is found, with a clear difference between -45 degrees for the front limbs and around -29 degrees for the hind limbs. The estimated confidence intervals of the LMM shows non-overlapping intervals between the front and hind limbs for the retraction angles while the intervals of the protraction angle are almost indifferent. On a cow level, the width of the boxes seems consistent for most cows and their limbs, which indicates that the variance is low, and the angles might be fairly consistent.

Cows show a similar protraction pattern as horses in walk, where the maximal protraction was reported an equal 19.6 degrees for both the front and hind legs. Retraction angles in horses differed between front and hind legs as well, but in contrast to cows a larger retraction has been observed for the hind legs (28.2 versus 23.0 degrees) [25]. This difference might be caused by the more sickle-hocked posture of the hind limbs in cows.

### Distal limb angle curve analysis

The distal limb angle curves were displayed both as median over all cows and steps, and as most typical curves per cow (Fig 7). The most typical distal limb angle curves were very similar between cows, as the cow specific curves overlap nicely, and the median curves show small MAD areas. The front and hind limbs show somewhat different patterns; the front limbs have a larger range (80 versus 60 degrees) and show a steeper decline just before and a steeper incline just after claw-off compared with the hind limbs.

In summary, the distal limb angle analyses show a larger retraction for the front limbs with a larger distal limb angle range and a steeper distal limb angle curve which might also explain the differences in stance duration between the front and hind limbs. A possible explanation might be the anatomical conformation of the front and hind limbs. The steeper decline just

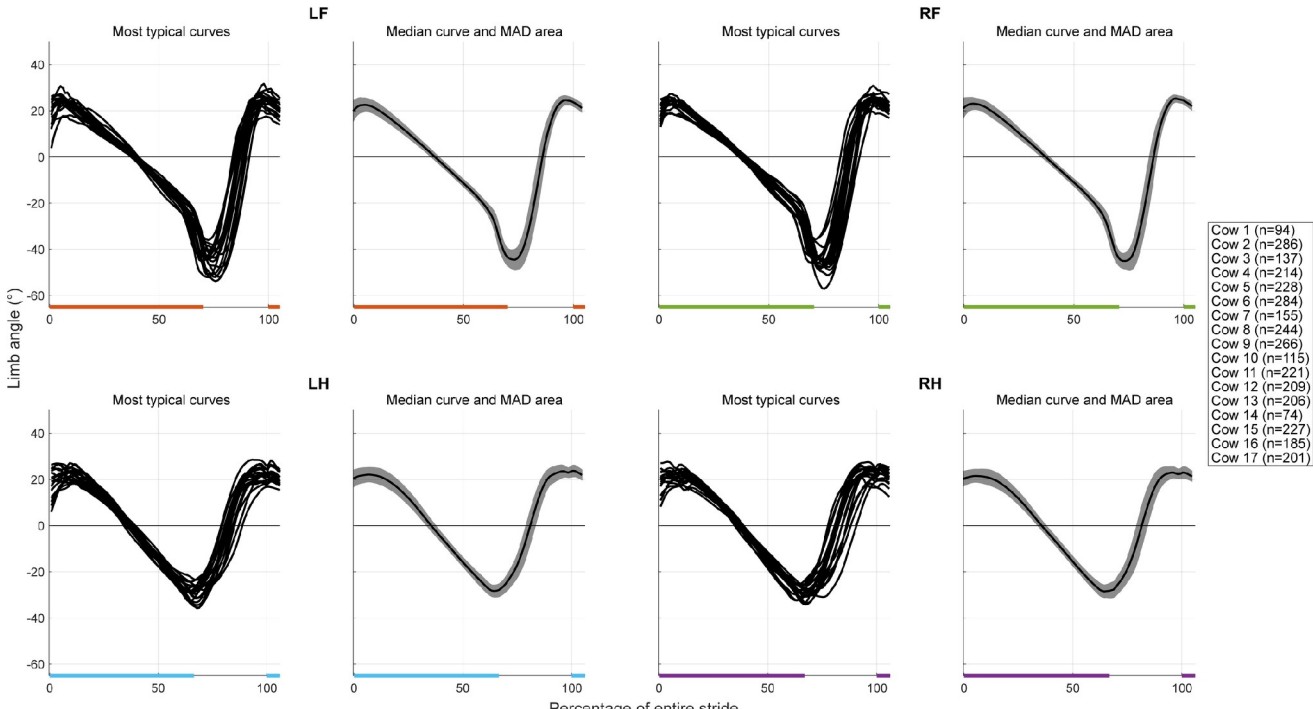

**Fig 7. Distal limb angle curves per limb.** For all limbs, the most typical curves per cow is shown on the left, and the median curve (black) with the MAD area (grey), calculated over all cows and steps, is shown on the right. The curves are shown on a scale from zero to 100% of the entire stride duration. The stance phases are indicated for every limb by the horizontal lines underneath the curves (orange: LF, green: RF, blue: LH, purple: RH).

before claw-off of the front limbs might be caused by flexion of the carpal joints just before full toe off. The steeper angle increase during swing phase of the front limbs follows from the similar swing phase duration for the hind and front limbs combined with a larger angle to cover.

## Vertical displacement analysis

The vertical displacement curves of the sacrum are shown in Fig 8, of the withers in S3 Fig and of the back in S4 Fig.

All the curves show one complete sinusoidal cycle per stride. For the sacrum (Fig 8), two peaks are found with the first peak around 25 and the second around 75% of the stride, which coincides with midstance of the LH limb and the RH limb respectively. This can be expected since the sacrum should be at its highest point during midstance of one of the hind limbs (Fig 9). For the withers (S3 Fig), the peaks around zero and 100% coincides with midstance of the RF limb and the peak around 50% coincides with midstance of the LF limb, which again can be anatomically expected. For the back (S4 Fig), the first peak is located just before zero, the second peak just before 50 and the third peak just before 100% of the stride, which happens to be just after the peaks of the sacrum and just before the peaks in of the withers. This can be explained by the attachment location on the 13[th] thoracic vertebra between the sacrum and the withers.

For the sacrum, most cows seem to have equal height peaks, which indicate a symmetrical gait pattern, and show a stable pattern, with small MAD area and very similar median and most typical curves. For the withers, the curves seem less symmetrical, indicated by less similar peak height and seem to show more variation between and within cows, indicated by less similar typical and median curves [37]. For the back, not all the cows seem to show three distinctive peaks and the curves are less smooth and show more variation, compared with the curves of the sacrum and the

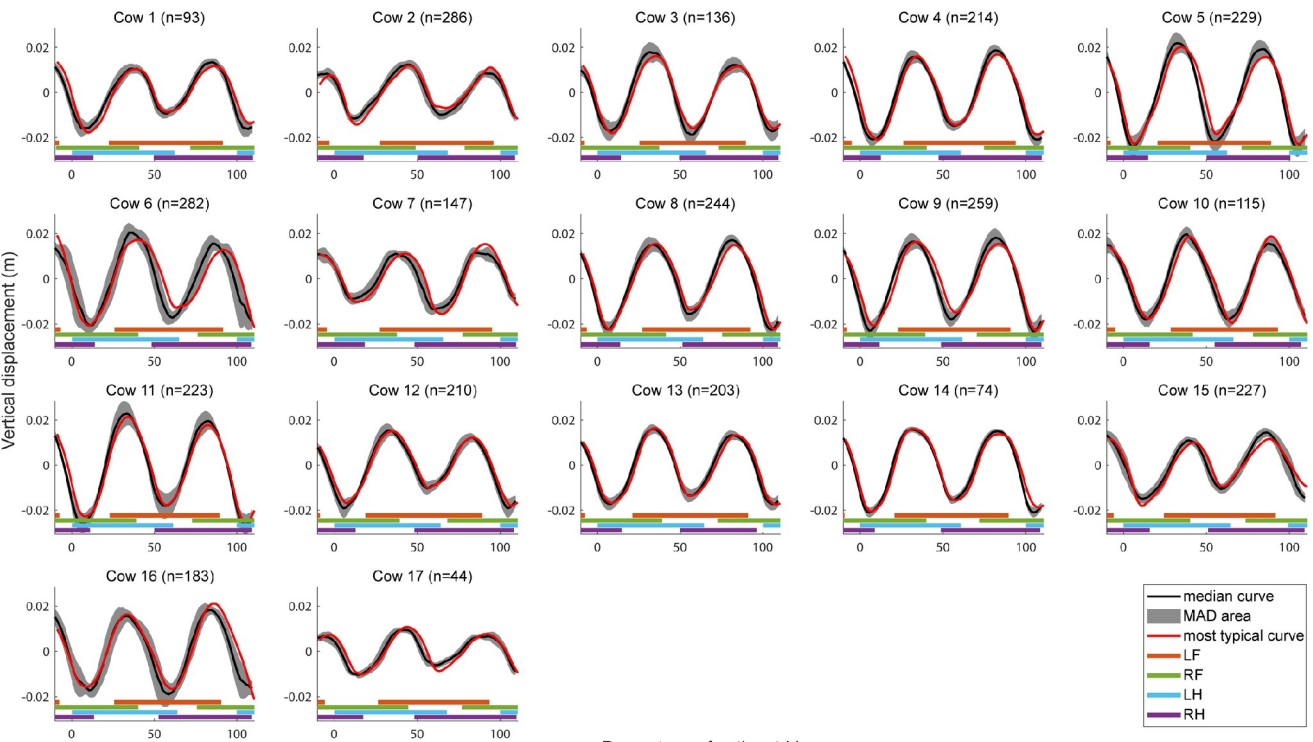

**Fig 8. Vertical displacement curves of the sacrum.** Per cow, the median curve (black), the MAD area (grey), and most typical curve (red) is shown on a scale from zero to 100% of the entire stride duration. The stance phases of the limbs are indicated by the horizontal lines underneath the curves (orange: LF, green: RF, blue: LH, purple: RH).

withers, which might be caused by the attachment of the sensor on the spinal process resulting in a rolling motion of the sensor over the spinal process in the more skinny cows.

The amplitude of the sacrum and withers curves is variable between 0.04 and 0.05 m in contrast to 0.02 and 0.03 m for the back.

The vertical displacement curves of the left (red) and right (green) tuber coxae seem to show one sinusoidal cycle per stride for each side in Fig 10. The peaks of the sinusoidal curve are located around 25 and 75% of the stride. For some cows, the locations of these peaks seem to happen a bit later during the stride, for example Cow 2, and are shifted towards 50 and 100% of the stride respectively. For the LTC, the first peak is the highest peak and coincides with midstance of the LH limb while the second and lowest peak coincides with midstance of the RH limb. The opposite is found for the RTC [38]. This finding seems obvious since the cow is pushing herself over the standing limb causing the hip to move over the standing limb and tilting the hip resulting in a higher vertical displacement for the side of the standing limb compared with the opposite side. All cows show a smooth and consistent pattern with a small MAD area around the median curve and the most typical curve seems almost similar to the median curve, except for Cow 7. The amplitude seems to be similar for all cows, around 0.1 m, except for Cow 7, which seems to have a flatter curve with small vertical differences between the two tuber coxae indication a low range of pelvic axial rotation. Overall, the curves seem smooth and similar between and within cows.

The vertical displacement curves of the head (S5 Fig) and collar (S6 Fig) show a less consistent sinusoidal pattern in about half of the cows for the head, and very unclear sinusoidal patterns for the collar, if present at all. For the cows with a sinusoidal pattern, the peaks are located

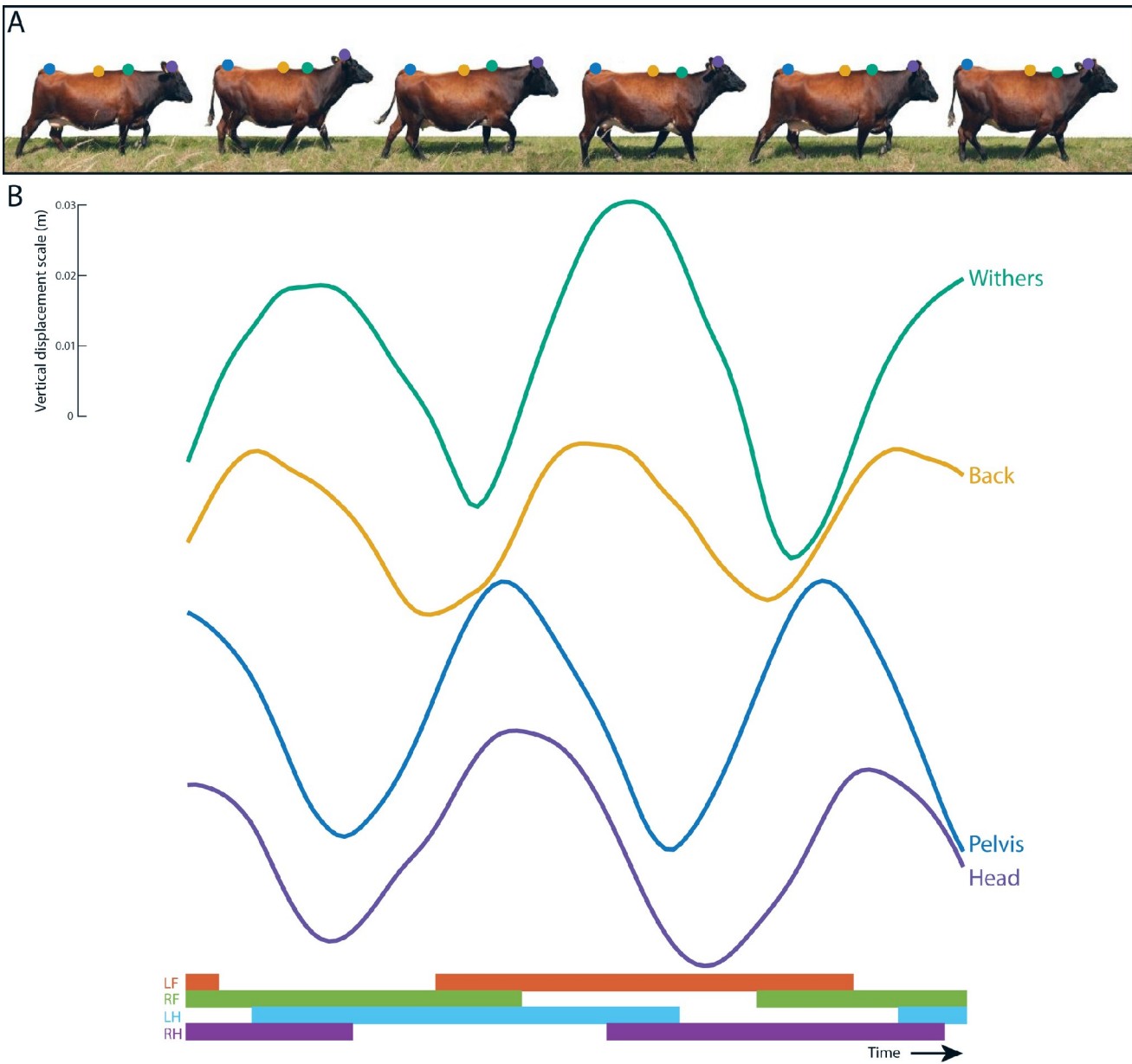

**Fig 9. Upper body displacement curve and footfall pattern of normal walking gait.** In A, different phases of a walking cow are shown with dots indicating the different sensor locations and their corresponding vertical signal pattern (green: withers, yellow: back, blue: sacrum and purple: head). In B, the median vertical displacement curves for the different sensor locations are shown from one cow, except for the head signal, which is selected from another cow for illustrative purposes. The synchronized footfall pattern is indicated underneath for all four limbs (orange: LF, green: RF, blue: LH and purple: RH). The scale of the signals is based on the true scale.

around zero, 50 and 100% of the stride. The peak around or just before 50% of the stride coincide with claw-off of the RF and claw-on of the LF limb. For the head, the sinusoidal patterns are less consistent than the patterns of the previous discussed locations and seem to differ more clearly between and within cows, with broader MAD areas and clear differences between the median and most typical curves for all cows. For the collar, the curves are even less consistent with even more variation within and between cows. The amplitude seems to differ between cows; between 0.02 and 0.04 m for the collar and between 0.02 and 0.1 m for the head.

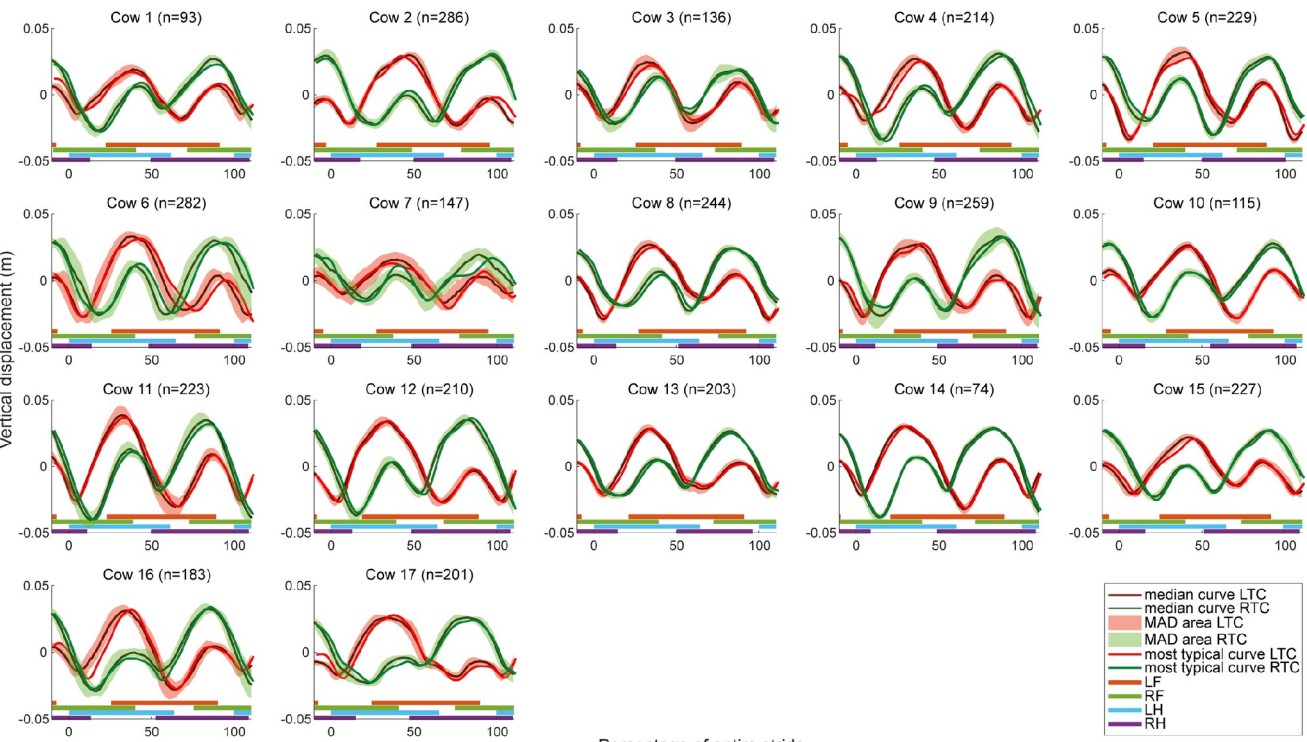

**Fig 10. Vertical displacement curves of the left and right tuber coxae.** Per cow, the median curve, MAD area and most typical curve is shown for the LTC (red) and RTC (green) on a scale from zero to 100% of the entire stride duration. The stance phases of the limbs are indicated by the horizontal lines underneath the curves (orange: LF, green: RF, blue: LH, purple: RH).

When comparing the curves of the different locations, it becomes clear that the sacrum, RTC and LTC show smooth sinusoidal curves without much variation between and within cows. The withers seem to have a bit more variation between and within cows although the sinusoidal patterns were still clearly visible. The back sensor seems to show slightly more variation between and within cows and the pattern seem to become less clearly visible. For the head and collar sensor, the curves seem to be less consistent and to show more variation between and within cows and for some cows, the sinusoidal pattern seem to have vanished for some cows completely.

An explanation for these differences might be that the cows were free to look around while walking resulting in head movements obscuring the stride related double sinusoidal vertical displacement curve. For the collar sensor, noise might have been introduced by the swinging movement of the sensor while walking because the collar was attached loosely around the cows' neck or because the sensor came loose during the measurement which happened in two measurements. Other signal processing technique might be more appropriate to evaluate the main frequency components from the noisy signal, for example using frequency component analysis methods such as Fourier series analysis [39–41].

Peaks were found around 25 and 75% of the stride for the sacrum, RTC and LTC, while the peaks of the withers were found around 0, 50 and 100%. This difference was also found in horses and the optimal phase shift observed in horses was found around 25% [42], which is in agreement with the phase shift found in this study. The phase shift of the vertical displacement curves between the different locations can be explained by the limb spread of the front and hind limbs [42]. In Fig 9, the footfalls and vertical displacement curves are illustrated of one

non-existing ideal cow to make this phenomena clearly visible. During midstance of the limbs, the body part above these limbs is at its highest point in contrast to the body part above the limbs in limb spread, which is at its lowest point. The phase shift between the front and hind part of the upper body can be explained by the asynchrony of the front and hind limbs at walk, where only one pair is in limb spread while one limb of the other pair is at midstance. The back sensor is located in between the front part (withers) and hind part (sacrum) of the upper body and therefore this curve shows peaks just before the withers and just after the sacrum [23]. The head is raised and lowered out of phase with the withers and is discussed as an energy saving mechanism of the walk which is seen in a majority of hoofed mammals [42].

The differences in height and depth of the peaks and valleys can be interpreted as an asymmetry of the vertical displacement. Such asymmetries are also described in horses and might be an indication of lameness [25], although asymmetries in vertical displacement of the sacrum and withers were also found in sound horses, in both walk and trot, and linked to differences in maximal protraction and retraction angles of the legs and motor laterality [37]. However, a sinusoidal pattern is needed to determine whether a symmetrical gait pattern is present and the absence of this, as in the head and collar curves, might therefore be problematic as indication for lameness using this approach. Nevertheless, modern signals analysis methods, including machine learning techniques, might be able to help to better further explore these complex signals in the near future.

The algorithm used for the claw-on and claw-off detection was developed and validated against the force plate and optical motion capture in horses [30]. No validation study was performed to evaluate the performance of this algorithm in cows since this involves letting the cows walk over a force plate, which was not possible. However, the signal appearance for cow and horse signals is quite similar and claw-on and claw-off detection were checked for consistence, as can be seen in S7 and S8 Figs. Temporal parameters might be affected when the claw-on and claw-off moments are not precisely detected, especially bipedal and tripedal support durations might be affected because these durations are short. It is not expected that the curve analysis of the upper body data is affected by the quality of claw-on and off detection since the curves are evaluated by adding a small interval around the claw-on and -off moments of the left hind limb and timing of footfalls is not a prerequisite for integration of the IMU data. Extreme outliers were manually removed when parameters were exceeding a threshold level, as described in the method section, to prevent these from obscuring our results.

## Conclusion

This is the first study that describes the kinematic gait characteristics of straight line walk in clinically sound dairy cows using body mounted IMUs at multiple anatomical locations. The method used in this study shows consistent results with low variance and speed normalization resulted in clearer differences between front and hind limbs for the stance duration. Furthermore, clear differences in distal limb angles between the front and hind limbs were found, as well as consistent and clear sinusoidal pattern for the vertical displacement curves of the tubera sacrale, withers, and left and right tuber coxae. For the head, back and collar sensors, signals with a sinusoidal pattern were found although they were less consistent and showed more variation between and within cows. These sensor locations are therefore less suitable for future exploration of lameness metrics in cows.

Even though the instrumentation used in this study might not be practical for daily implementation on farms, it allowed us to explore in unprecedented detail the kinematic gait characteristics of cows at walk. Future analysis of these signals in lame cows will allow us to identify the best features that can be used from IMU data to objectively quantify lameness and might

be useful for the development of an automatic recognition method and extensions to computer vision techniques.

## Supporting information

**S1 Fig. Schematic representation of analysis steps performed.**
(TIF)

**S2 Fig. Definition of distal limb angles.** Maximal protraction is the maximal forward protraction (positive angle) from midstance and maximal retraction is the maximal backward retraction (negative angle) from midstance measured at the metacarpus/-tarsus in the sagittal plane, as adapted from horses [28, 32].
(TIF)

**S3 Fig. Vertical displacement curves of the withers.** Per cow, the median curve (black), the MAD area (grey), and the most typical curve (red) is shown on a scale from zero to 100% of the entire stride duration. The stance phases of the limbs are indicated by the horizontal lines underneath the curves (orange: LF, green: RF, blue: LH, purple: RH).
(TIF)

**S4 Fig. Vertical displacement curves of the back.** Per cow, the median curve (black), the MAD area (grey), and most typical curve (red) is shown on a scale from zero to 100% of the entire stride duration. The stance phases of the limbs are indicated by the horizontal lines underneath the curves (orange: LF, green: RF, blue: LH, purple: RH).
(TIF)

**S5 Fig. Vertical displacement curves of the head.** Per cow, the median curve (black), the MAD area (grey), and the most typical curve (red) is shown on a scale from zero to 100% of the entire stride duration. The stance phases of the limbs are indicated by the horizontal lines underneath the curves (orange: LF, green: RF, blue: LH, purple: RH).
(TIF)

**S6 Fig. Vertical displacement curves of the collar.** Per cow, the median curve (black), the MAD area (grey), and the most typical curve (red) is shown on a scale from zero to 100% of the entire stride duration. The stance phases of the limbs are indicated by the horizontal lines underneath the curves (orange: LF, green: RF, blue: LH, purple: RH).
(TIF)

**S7 Fig. Claw-on and claw-off detection relative to the acceleration signal along horizontal axis.** Raw acceleration data of the LF limb of cow 16 was used to show the claw-on (red) and claw-off (green) detections.
(TIF)

**S8 Fig. Claw-on and claw-off detection relative to the gyroscope signal along the left-right axis.** Raw gyroscope data of the LF limb of cow 16 was used to show claw-on (red) and claw-off (green) detections.
(TIF)

**S1 File. Temporal parameters used in this study.** Excel file with stance duration, speed normalized stance duration, bipedal and tripedal support durations, speed normalized bipedal and tripedal support durations, maximal protraction and retraction angles of the distal limbs and cows.
(XLS)

**S1 Table. Cow characteristics.** (SR stands for the Swedish Red-and SH stands for the Swedish Holstein breed).
(XLSX)

## Acknowledgments

We acknowledge that cooperation of the cows was vital to collect these data. Furthermore, we would like to thank J. Lundblad, C. Frisk and the animal caretakers of the Swedish Livestock Research Centre Lövsta for their contribution to the data collection. And at last, J. van den Broek for statistical guidance.

## Author Contributions

**Conceptualization:** M. Rhodin, P. H. Andersen, E. Hernlund.

**Data curation:** M. Tijssen.

**Formal analysis:** M. Tijssen, F. M. Serra Bragança.

**Funding acquisition:** M. Rhodin, P. H. Andersen, E. Hernlund.

**Investigation:** M. Tijssen, K. Ask, M. Rhodin, E. Telezhenko, C. Bergsten, E. Hernlund.

**Methodology:** M. Tijssen, K. Ask, M. Rhodin, P. H. Andersen, E. Telezhenko, C. Bergsten, E. Hernlund.

**Project administration:** K. Ask, M. Rhodin.

**Resources:** K. Ask, E. Telezhenko, C. Bergsten.

**Software:** M. Tijssen, F. M. Serra Bragança.

**Supervision:** F. M. Serra Bragança, M. Rhodin, M. Nielen, E. Hernlund.

**Visualization:** M. Tijssen.

**Writing – original draft:** M. Tijssen, K. Ask.

**Writing – review & editing:** F. M. Serra Bragança, K. Ask, M. Rhodin, P. H. Andersen, M. Nielen, E. Hernlund.

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
