## [Decision Letter · Decision Letter 0]

25 Feb 2021

PONE-D-20-38036

Biomechanical gait characteristics of straight line walk in clinically sound dairy cows

PLOS ONE

Dear Dr. Tijssen,

Thank you for submitting your manuscript to PLOS ONE. After careful consideration, we feel that it has merit but does not fully meet PLOS ONE’s publication criteria as it currently stands. Therefore, we invite you to submit a revised version of the manuscript that addresses the points raised during the review process.

Thank you for this well-written manuscript. Both expert reviewers and I found it very interesting. However, the reviewers have identified some areas where the manuscript needs to be improved and I concur with their view. Please carefully address all the reviewers' comments when/if revising your manuscript.

We look forward to receiving your revised manuscript.

Kind regards,

Angel Abuelo, DVM, MRes, MSc, PhD, DABVP (Dairy), DECBHM

Academic Editor

PLOS ONE

Journal Requirements:

3. We note you have included a table to which you do not refer in the text of your manuscript. Please ensure that you refer to Table 2 in your text; if accepted, production will need this reference to link the reader to the Table.

Reviewers' comments:

Reviewer's Responses to Questions

**Comments to the Author**

1. Is the manuscript technically sound, and do the data support the conclusions?

Reviewer #1: Partly

Reviewer #2: Partly

2. Has the statistical analysis been performed appropriately and rigorously? 

Reviewer #1: Yes

Reviewer #2: I Don't Know

3. Have the authors made all data underlying the findings in their manuscript fully available?

Reviewer #1: No

Reviewer #2: Yes

4. Is the manuscript presented in an intelligible fashion and written in standard English?

Reviewer #1: Yes

Reviewer #2: Yes

5. Review Comments to the Author

Reviewer #1: Dear authors,

With great interest I have read your manuscript and below you will find my comments.

The article describes a methodology to assess the locomotion of cows. IMU’s attached to eleven anatomical landmarks were used to show their motion and orientation paths relative to gravitational force (angle) during steps and strides of cows at walk. Up front these cows were checked for their locomotory health and during the experiment guided in forward walking in a straight line at voluntary speed. It is assumed the IMU’s accurately describe the motion of the underlying structure of the musculoskeletal system. The aims are 1) to provide a “biomechanical description of sound walking”, to later allow for anomaly detection that might be related to lameness in ruminants, AND 2) find the best anatomical location using IMU’s for lameness detection.

The authors provide an overwhelming amount of figures, curves, table, etc. in which is shown the intra- and intercow variation in sound walking is significant. Luckily the normalization of strides and steps data can be decently done using the describes statistics. Hence, sensor output of the IMU’s can be used for gait analysis. However, an underlying “biomechanical model” is lacking. Front- and hindlimb anatomy and their subsequent functioning in walking, when propelling the animal forward, is different. The, in my opinion oversimplification of, protraction and retraction limb angles measured by the 4 IMU’s attached to the metatarsus and metacarpus, is neglecting the intralimb coordination to be able to describe full-limb biomechanics. The spatial characteristics of these two particular body segments should be better explained and preferably validated in such a model.

The temporal aspects of limbs’ stance and swing phases at walk can be recorded, but the authors fail to provide a validation, except for the citations that it works in horses. But from a hooved animal to a cloven-hooved animal this might need additional work, in the light of a lift-off or push-off. Not to forget lameness origin is mainly located at lateral hind claws (possibly resulting in more sickle-hocked walking, hence sagittal plane analysis might not be sufficient). Nothing as such is discussed.

Despite the symmetry of motion and orientation paths in the cows’ steps, strides and mentioned support phases shows promise to describe “normal” locomotion patterns, bottom line is; the manuscript is about a sensor technology that measures data, which is shown in all kinds of descriptions, visualizations (average pathways, etc) and heavy statistics, but only assuming a relation with walking. There is no test of showing part of the data falls within the MAD’s of the other part (train and testset-approach) nor a validation using an (gold) standard technology (e.g. Qualisys system). That leaves me as reader with the opinion what I already knew; IMU’s can be used for symmetry assessment in gait analysis (first aim). In addition the second aim, best anatomical landmark for lameness detection is not specifically addressed in the discussion nor conclusion.

So I encourage you to provide a better context and reconsider what specific research questions that can be addressed using this dataset. In addition I would like to read a more in-depth discussion.

Major concerns

The introduction builds a clear case from clinical perspective it is a logical continuation of research to start using IMU’s for lameness detection. However, there is a huge development going on in machine vision and deep learning approaches for lameness detection in cattle (and horses), which in theory could do similar analysis, but then without the hassle of mounting sensors on animals. Neglecting this part of the scientific domain in the introduction seems inappropriate. I suggest the authors provide clear reasoning why their approach is worthwhile pursuing, relative to artificial intelligence, AND the differences between the clinical practice of horses and cows. I can imagine the technology is preferred for its quality, but options for cows in a dairy practice seems very limited.

Line 339: “80 versus 60 degrees”. So what is the case? Because with equal limb length, a wider angle (Range of Motion), simply means increased step lengths. That's obviously NOT the case, as all limbs have similar step-lengths. Due to the lack of an animal model there is no clear explanation. The only way to explain, the difference is in the metatarsus/-carpus orientation and motions relative to the adjacent segments in the corresponding limbs. Since these have not been recorded, it is inappropriate to claim more than "the technology is able to detect deviations in symmetry" with respect to the pathway and orientation of the metatarsus/carpus and subsequent bi-, tripedal support phases.

Line 474-475: I assume you’ve deliberately choose not to use a force plate. Why was this not possible? The video that was synchronized with one of the IMU’s could be useful in this.

Conclusion is too much a continuation of the discussion. Please only mention the conclusions that can be derived form your results.

Minor comments:

Line 96-97: claws were trimmed 1 to 90 days prior to the experiment, can you be more specific?

Line 111; “Sensors attached to skin”, nothing is discussed about skin displacement relative to the musculoskeletal system and potential motion artifacts.

Line 124: “calibrated”, seems more like the sensor was tared for midstance, relative to the gravitational force (angle = 0˚).

Line 132: “corridor of 72 meter”, no beams, fences or cubicles next to it?

Line 134-136: This seems in conflict, self-chosen pace versus "encouraging", that might have caused some head turns at least. Any effect on withers, neck and head sensors?

Line 144-157: I might not understand the process, but I prefer to read this in a (chrono)logical order. primary selection:

1) video observation; straight line

2) Physical handling? Not clear what is meant with this.

Data processing => footfall, basically the metatarsus/carpus pathway's. final selection:

3) checked for regularity, at least four strides (and not necessarily adjacent to one another?).

I suggest to create a flowchart that shows the data selection process till the analysis, to clarify things.

Line 159: “the selected IMU data” this is all data or only the selected strides?

Line 161-169; A brief explanation of stride segmentation would be useful. In addition, to what extent is an orthogonal projection (to the sagittal plane) an allowed simplification?

Line 183-185; it is not clear what amount of limbs (pairs) was used. Suggest to make this clear.

Line 185: “bootstrapping” is a 17 cows sample good enough to estimate the extremes of the populations distribution? The process could be better described.

Line 227-228: What caused the unequal number of strides? Was the sensor not recording a claw-on or claw-off? It’s clear that it was observed, but authors should provide a (potential) reason for this.

Line 437-439: If something is/seems more appropriate, then why is this not used?

Line 454: “and is discussed”, interesting to know the opinion of the authors.

Reviewer #2: Please find my comments enclosed to the manuscript PONE-D-20-38036, entitled: Biomechanical gait characteristics of straight line walk in clinically sound dairy cows.

The authors provided interested data. However, some Revision are needed before publishing.

6. PLOS authors have the option to publish the peer review history of their article (what does this mean?). If published, this will include your full peer review and any attached files.

Reviewer #1: **Yes: **PPJ (Rik) van der Tol

Reviewer #2: No

---

## [Author Response · Author response to Decision Letter 0]

16 Apr 2021

Journal Requirements:

The right funding statement is: “This research was funded by Swedish Research Council FORMAS (http://www.formas.se), grant number 2016-01760 (MR) and 2018-00737 (EH).”

3. We note you have included a table to which you do not refer in the text of your manuscript. Please ensure that you refer to Table 2 in your text; if accepted, production will need this reference to link the reader to the Table.

Reviewers' comments:

Reviewer #1:

Dear authors,

With great interest I have read your manuscript and below you will find my comments.

Dear P.P.J. van der Tol,

Thank you for your comments and extensive feedback! We have tried to revise the manuscript according to your recommendations and feel that this has improved the work a great deal. To be sure to respond on all the points made and not miss any, we have added some numbers to your letter.

The article describes a methodology to assess the locomotion of cows. IMU’s attached to eleven anatomical landmarks were used to show their motion and orientation paths relative to gravitational force (angle) during steps and strides of cows at walk. Up front these cows were checked for their locomotory health and during the experiment guided in forward walking in a straight line at voluntary speed. It is assumed the IMU’s accurately describe the motion of the underlying structure of the musculoskeletal system. The aims are 1) to provide a “biomechanical description of sound walking”, to later allow for anomaly detection that might be related to lameness in ruminants, AND 2) find the best anatomical location using IMU’s for lameness detection.

This is indeed correct.

For our purpose to describe healthy walking patters in cattle, we deployed a well-developed and extensively used horse method (Bosch et al. 2018). We hypothesized that the analogy between horses and cows was good enough to adapt these methods. Clearly, there are anatomical differences between horses and cows, such as the differences in evolutionary digital reduction of the manus /pedis, the well-developed proximal musculature in the horse, the larger abdominal volume of the cow and the functional anatomy of the back. We did not elaborate on these differences in the introduction, nor did we account for these differences in our hypothesis. We now have added a couple of sentences in the introduction to highlight this a bit more. However, it was not our goal to describe specific differences between horses and cows. We have also adapter the algorithms developed for the horses in order to extract the relevant features from our data and to account for differences in locomotion between the 2 species.

Our future goal is to compare walking pattern of lame cows with these healthy patterns to distinguish the changes in kinematic gait characteristics and use this knowledge as a foundation for automatic recognition method (based on video detection). In order to further advance any machine learning or computer vision methods, this work needs to be done to build a solid foundation of knowledge. We have elaborated a bit more on this in the conclusion section where we now also describe a future outlook. However, this will be more thoroughly discussed in our next paper which aims to determine the most informative biomechanical gait characteristic and sensor location for the identification of induced hindlimb lameness in cows during straight line walk.

The authors provide an overwhelming number of figures, curves, table, etc. in which is shown the intra- and inter cow variation in sound walking is significant. Luckily the normalization of strides and steps data can be decently done using the describes statistics. Hence, sensor output of the IMU’s can be used for gait analysis. However, an underlying “biomechanical model” is lacking (1). Front- and hindlimb anatomy and their subsequent functioning in walking, when propelling the animal forward, is different. The, in my opinion oversimplification of, protraction and retraction limb angles measured by the 4 IMU’s attached to the metatarsus and metacarpus, is neglecting the intralimb coordination (2) to be able to describe full-limb biomechanics (3). The spatial characteristics of these two particular body segments should be better explained and preferably validated in such a model.

(1): We want to express our gratitude for your extensive feedback and suggestions. We display the kinematic signal pattern and time discrete variables of the walking cows. We have not extended the interpretation of the data into a “biomechanical model” and it was not really clear to us what type of model is sought. We agree that developing a model to explain or to describe cow movements might be very interesting, especially when you are able to validate such a model. However, this was not the scope of this study. The goal of this paper was to explore and describe the healthy walking pattern of dairy cattle using body mounted IMUs.

We agree that the title might also give the wrong impression and is not really clearly describing what we are doing. Therefore, we changed the word “biomechanical” to “kinematic”.

(2): Interlimb coordination is described in this paper indirectly by analyzing the stance duration, next to bipedal and tripedal support durations and limb angles. We did not look at limb-coordination per se (e.g. by using cross correlation) because to our best knowledge, there has been no proven value of this in horses nor in cows.

(3): We agree that we made an unfortunate over-simplification of the limb protraction and retraction angles. We have tried to make it more clear in the manuscript that it was the motion of the metatarsal / metacarpal segments (the distal limb –not the entire limb) that were measured and nothing else.

The temporal aspects of limbs’ stance and swing phases at walk can be recorded, but the authors fail to provide a validation, except for the citations that it works in horses. But from a hooved animal to a cloven-hooved animal this might need additional work (4), in the light of a lift-off or push-off. Not to forget lameness origin is mainly located at lateral hind claws (possibly resulting in more sickle-hocked walking, hence sagittal plane analysis might not be sufficient) (5). Nothing as such is discussed.

(4): The detection of claw-on/off timing was based on the algorithm we developed for horses (Braganca et al. 2017). Although the algorithm has been previously described, it was not our goal with the current manuscript to fully disclose the complete data analysts process. As an initial step in our data analysis, we have investigated the signals (gyroscope and accelerometer) generated from our IMUs and the gait event detected which can be found in the S3 and S4 Figures. We agree that in order to fully establish accuracy and precision of these detections we would need further work with force plates for example – however, that is not feasible and was not the aim of this manuscript. Nevertheless, we have added this point as a limitation of our current work.

(5): We are grateful for this suggestion. Off course there is more data to study and evaluate, for example lateral claw unloading in transverse plane. However, we made a selection from the huge amount kinematic data that we have collected based on the knowledge that is obtained from horses. All the cows used for this study were selected based on normal limb conformation and straight walking manner to avoid measuring cow hocked or sickle hocked walking. Furthermore, we did not describe that this posture might lead to a higher incidence of hind limb lameness, although in our next paper we will discuss this more thoroughly and describe adaptation strategies for hindlimb lameness in more detail.

Despite the symmetry of motion and orientation paths in the cows’ steps, strides and mentioned support phases shows promise to describe “normal” locomotion patterns, bottom line is; the manuscript is about a sensor technology that measures data, which is shown in all kinds of descriptions, visualizations (average pathways, etc.) and heavy statistics, but only assuming a relation with walking. There is no test of showing part of the data falls within the MAD’s of the other part (train and test set-approach) nor a validation using an (gold) standard technology (e.g. Qualisys system) (6). That leaves me as reader with the opinion what I already knew; IMU’s can be used for symmetry assessment in gait analysis (first aim) (7). In addition, the second aim, best anatomical landmark for lameness detection is not specifically addressed in the discussion nor conclusion (8).

(6): In this study, we did not perform any validation with a motion capture system or force plate since this was not feasible to perform on a dairy farm. However, these validation studies are extensively performed in horses (Braganca et al. 2017, Bosch et al. 2018) and we hypothesized, as described before, that the method was adaptable to cows without performing these validations studies again. Furthermore, we also implemented this method on lame cows which will be described in our next paper (in preparation). For this project, we had to keep in mind that we wanted to perform extensive biomechanical research based on a well-developed method, but it had to be feasible and applicable on a research dairy farm.

(7): Although, these type of studies were previously performed in horses and we hypothesized that this method was adaptable to cows, this is still the first study that used IMUs to assess symmetry in walking in dairy cattle.

(8): The second aim to describe which anatomical attachment locations are suitable for lameness detection is discussed in Line 475-479. In this section, we described that the head and collar locations might be less suitable for lameness detection due to the less clear sinusoidal pattern of the signal. However, it might be possible to use these locations when these signals are analyzed with different types of analysis techniques, such as machine learning, which were beyond the scope of this study. This is also described in the conclusion section (line500-502). We have changed the wordings of this section to make clear that this is our conclusion for the second aim.

So, I encourage you to provide a better context and reconsider what specific research questions that can be addressed using this dataset. In addition, I would like to read a more in-depth discussion.

We have provided the manuscript with track changes and hope that we have succeeded in providing a better context.

Major concerns

The introduction builds a clear case from clinical perspective it is a logical continuation of research to start using IMU’s for lameness detection. However, there is a huge development going on in machine vision and deep learning approaches for lameness detection in cattle (and horses), which in theory could do similar analysis, but then without the hassle of mounting sensors on animals. Neglecting this part of the scientific domain in the introduction seems inappropriate. I suggest the authors provide clear reasoning why their approach is worthwhile pursuing, relative to artificial intelligence, AND the differences between the clinical practice of horses and cows. I can imagine the technology is preferred for its quality, but options for cows in a dairy practice seems very limited.

We hope that we have provided a clear answer on this in the previous answers.

Line 339: “80 versus 60 degrees” (9). So, what is the case? Because with equal limb length, a wider angle (Range of Motion), simply means increased step lengths. That's obviously NOT the case, as all limbs have similar step-lengths. Due to the lack of an animal model there is no clear explanation. The only way to explain, the difference is in the metatarsus/-carpus orientation and motions relative to the adjacent segments in the corresponding limbs. Since these have not been recorded, it is inappropriate to claim more than "the technology is able to detect deviations in symmetry" with respect to the pathway and orientation of the metatarsus/carpus and subsequent bi-, tripedal support phases (10).

(9): The anatomical differences, such a more sickle-hocked posture of the hind limbs and difference in limb length between the front and hind limbs, results in a difference in range of the limb angles (80 vs 60 degrees). We describe this very briefly in Line 332-333 and Line 346-352. The limb angles are measured at the metacarpus/-tarsal segment of the limbs which was not clearly stated in the text and the Table 1. We have changed this now. Since, the maximal protraction and retraction angles are measured at the metacarpus/-tarsus segment, it might be that we overestimated the retraction angles due to the flexion of the carpal joints just after toe-off (described in Line 349-352).

(10): The IMU sensor system used in this study has been extensively used in equine locomotion research and has shown to detect very accurately motion asymmetry. We also detect in cows deviations in symmetry on such a very fine level that we are not sure whether this is clinically relevant. Furthermore, this system is validated against the gold standard of both kinetics (force plate) and kinematics (motion capture system) (Braganca et al. 2017).

I hope that this answer clarifies our consideration more clearly and I have addressed the minor concerns pointwise down below.

Line 474-475: I assume you’ve deliberately chosen not to use a force plate. Why was this not possible? The video that was synchronized with one of the IMU’s could be useful in this.

Indeed, we did not use a force plate during this experiment because we wanted to describe the kinematics of the walking pattern of dairy cattle. Force plates give kinetic information which was not the information we deemed necessary for this purpose. However, the algorithm that we used for claw-on and claw-off detection was validated against the force plate and motion capture system in horses. Furthermore, we compared the signal appearance of the cows with that of the horses and evaluated the performance of the algorithm used for this purpose. We found no major errors and we expected that the algorithm performed sufficient enough to measure the temporal parameters with a resolution high enough to detect changes due to lameness induction (performed in our next paper).

Conclusion is too much a continuation of the discussion. Please only mention the conclusions that can be derived from your results.

I have changed the wording of this section to make clear that the conclusion is not a continuation of the discussion.

Minor comments:

Line 96-97: claws were trimmed 1 to 90 days prior to the experiment, can you be more specific?

The trimming dates for every cow can be found in S1 Table.

Line 111; “Sensors attached to skin”, nothing is discussed about skin displacement relative to the musculoskeletal system and potential motion artifacts.

Artefacts due to skin displacement was already investigated in horses by van Weeren et al. (van Weeren, van den Bogert, and Barneveld 1990) and we had no reason to believe that this was different in cows.

Line 124: “calibrated”, seems more like the sensor was tared for midstance, relative to the gravitational force (angle = 0˚).

The sensors were calibrated to identify the offset of the gyroscope and to calibrate the gravity vector. During the preprocessing, a correction was performed when an offset was found. The sensors were not tared for midstance during this process.

Line 132: “corridor of 72 meter”, no beams, fences or cubicles next to it?

Indeed, there were cubicles and fences on both sides of the corridor and I now added this in the description (line 139).

Line 134-136: This seems in conflict, self-chosen pace versus "encouraging", that might have caused some head turns at least. Any effect on withers, neck and head sensors?

Thank you for reading so carefully, indeed the chosen wording became contracting. I therefore removed “to encourage to walk straight continuously” to make it clear that we did let the cows walk their own pace. I selected the intervals were the cow did not look behind her and walked in very relaxed manners straight forward. So, we cannot see any head turns in the data selected for this paper.

Line 144-157: I might not understand the process, but I prefer to read this in a (chrono)logical order. primary selection: 1) video observation; straight line, 2) Physical handling? Not clear what is meant with this. Data processing => footfall, basically the metatarsus/carpus pathway's. final selection: 3) checked for regularity, at least four strides (and not necessarily adjacent to one another?). I suggest creating a flowchart that shows the data selection process till the analysis, to clarify things.

Thank you for your feedback. We agree that a lot of steps are performed to finally came to the analyzed data for this paper. We have therefore added a flowchart to the supplementary materials (S1 Figure) and I hope that this makes the workflow clearer.

Line 159: “the selected IMU data” this is all data or only the selected strides?

Indeed, we mean the selected intervals described in the previous paragraph. To make this clearer, we have changed “the selected IMU data” into “the selected intervals from the IMU data” (line 165).

Line 161-169; A brief explanation of stride segmentation would be useful. In addition, to what extent is an orthogonal projection (to the sagittal plane) an allowed simplification?

Stride segmentation was performed based on the impact moment (claw-on) of the left hind limb. We have added this description and reference to the article in which this method is described in line 168-169.

Line 183-185; it is not clear what number of limbs (pairs) was used. Suggest to make this clear.

Thank you for noticing. We agree that the meaning of limb (pairs) is not clear because it was not mentioned before. Therefore, we added “Stance duration was calculated for all limbs separately while bipedal and tripedal support durations were calculated for all possible 2 or 3 limb pairs combinations. The definitions of these durations can be found Table 1.” to line 187-188.

Line 185: “bootstrapping” is a 17 cows sample good enough to estimate the extremes of the populations distribution? The process could be better described.

We used bootstrapping to calculate the median and the 95% confidence intervals of each parameters to avoid any effects of the low number of observations for the cows with less observations then others.

Line 227-228: What caused the unequal number of strides? Was the sensor not recording a claw-on or claw-off? It’s clear that it was observed, but authors should provide a (potential) reason for this.

Thank you for noticing this. We selected only complete stride cycles to be sure that stance durations could be normalized with the stride duration of the left hind limb. To do this, we needed the same number of steps from all limbs and therefore we only selected complete stride cycles (from LH claw on to the next LH claw on). However, the cows were walking until the end of the corridor and then turn around which sometimes happened when the stride cycle was not completed yet. For example, the right front limb was not placed on the ground yet or that only the left hind limb made a step of a new stride cycle. So, we had to exclude these steps from analysis.

Line 437-439: If something is/seems more appropriate, then why is this not used?

In this paper, we used the time domain signals and the parameters that show high potential for lameness detection in horses. Furthermore, our future goal is to develop a system based on computer vision that detects changes in the walking pattern of cows over time. Therefore, frequency domain analysis seems not straight forward to use. However, we agree that these signals might contain very useful information and we wanted to highlight this and opt for other analysis procedures that can be performed in the future such as frequency domain analysis and machine learning techniques.

Line 454: “and is discussed”, interesting to know the opinion of the authors.

We are sorry to disappoint but we cannot find this statement in line 454. The only time we “discussed” a statement was in line 458 where we described the timing of the head nod which happened out of phase with the movement of the withers as discussed in the article of Loscher et al. (Loscher et al. 2016) (ref 42 of the manuscript).

Reviewer #2:

With interest I read your paper. The paper provides relevant data regarding using IMU sensors to detect diverse kinematics variables at multiple anatomical locations in healthy cows. Well written! 

Below you find my minor comments

Thank you for your careful consideration of this paper and for your feedback. Below I have written my response for every point, I hope this will clarify all the changes and considerations we made.

Abstract:

No values available in the abstract. This makes the abstract more "descriptive"

I removed the first sentence of this paragraph to make the abstract more descriptive.

36-38: I suggest revising the sentence as you so far not performed your measurements in lame cows. Delete "promising" and add "maybe improve automated lameness detection". See lines 76-78.

Thank you for commenting on this. I changed the wording in line 36.

Keywords:

What is the difference between the pattern and characteristics?

When speaking in signal processing terms: q pattern is something like a shape patten (over time) or a pattern in a signal (like a square wave, sinusoid etc.), while a characteristic is something that defines a signal, like the amplitude or specific frequency content or a specific feature, happening at one time point, that can be detected from the signal.

Introduction:

40-41: add reference. Many biomechanical studies are so far performed and aimed to detect lameness in cattle.

I have added a reference to this statement.

55-56: not really. Check the PubMed. Several works have described the biomechanical gait in cattle. E.g. Flower et al. 2005 

Thank you for your feedback. I agree that “only a couple studies” is not a good statement and that its better to call it “several works” and I also mentioned the Flower et al. paper (ref 17).

65: temporal stride parameters. Why not kinematics parameters instead of temporal parameter? 

Kinematic parameters are all parameters we did evaluate in this paper and we want to highlight that we evaluated different kinds of kinematic parameters such as vertical displacement, limb angles and temporal parameters (e.g. stance and support durations). We hope that this explanation makes the difference between kinematic and temporal parameters clearer.

72-73: how the authors can explain the importance of applying the sensors in upper body in comparison to the "traditional" limb positions?

The importance of the upper body sensor locations is already descripted in horses in the paper of Serra Braganca et al (Serra Braganca, F. M., et al. (2020). "Adaptation strategies of horses with induced forelimb lameness walking on a treadmill." Equine Vet Journal.). The upper body moves slower relative to the limbs and therefore asymmetry is more easily assessed by the eye. The upper body motion alterations can be of particular interest for future computer vision surveillance techniques since it is more visible to wall mounted cameras and less prone to occlusion from fellow herd mates. 

Materials and Methods:

84-85 were the cows guided or the halter broken, and the cows walked freely. What is happen when the cows running or when there were signal artifacts of IMU?

The cows were free to walk in their own pace and they were not guided with a halter. The cows were not running because we did not push or scare them to move forward because we wanted to measure a natural walking flow with stride-by-stride regularity. We removed the intervals for analysis where the cows showed behavior like kicking with the limbs or when the cow stopped walking.

97-98: must clearly defined. What about SU or other hoof diseases?

We changed the wording in the method section to “during which no clinically significant claw disorders were recorded” in order to clearly state the selection criteria. We considered what was clinically significant from the claw trimming records and gave two examples of disorders that were considered clinically neglectable. However, this became not clear in the way we wrote it before.

141: you used a low frame for you held-camera. How you performed the synchronization with IMU? It needs at least high-speed camera for this synchronization

We only used the synchronization with the handheld camera to watch the videos and determine the intervals with stride-by-stride regularity. This was performed with the naked eye and no high-speed camera was needed for this purpose. We have added a flowchart to the supplementary materials to make to selection process clearer, we hope this will answer your question.

193: may you can add the unites to the Table 1. It is not more realistic to consider the relative measurements (%) rather than the absolute values (s)?

Thank you for noticing, we have added the units of the parameters to Table 1. In this study, we have evaluated both absolute and relative measurement (fraction of entire stride duration, also called duty factor in horses) because we wanted to see if it was indeed more realistic to only consider relative measurements. This is also discussed later in the discussion Line 305-307 (“Walking speed normalization improved illustration of stance duration patterns between the front and hind limbs although for the bipedal and tripedal support durations, less effect was observed.”)

Results and Discussion

224-225: may you can add the median

We have now added the median of the selected strides.

228: Figure 2 and all the figures have poor resolution, so it is difficult to follow them.

Thank you for noticing. We used the PACE tool as described and all the figures were transposed with a good resolution. We will check with the academic editor to make sure that the figures have a good quality.

Have the authors checked the differences of temporal events (e.g. stance phase duration) between left and right limbs? How are the breed, DIM, lactation numbers etc. affecting your results?

We evaluated the temporal parameters for all cows and all limb (pairs) separately and if differences were found we described them in the discussion section. However, we did not perform statistical analysis to determined the effects of breed, DIM and lactation number on the parameters because we did not expect to find differences based on these parameters due to our selection criteria (described in Line95-101) and limited sample size in this experimental set up. The goal of our manuscript was not to look into these differences but to account for these by including several subjects. In our analysts, Animal ID is used as a random effect exactly to exclude some of these differences that might exist. 

References

Bosch, S., F. Serra Braganca, M. Marin-Perianu, R. Marin-Perianu, B. J. van der Zwaag, J. Voskamp, W. Back, R. van Weeren, and P. Havinga. 2018. "EquiMoves: A Wireless Networked Inertial Measurement System for Objective Examination of Horse Gait." Sensors (Basel) 18 (3). doi: 10.3390/s18030850.

Braganca, F. M., S. Bosch, J. P. Voskamp, M. Marin-Perianu, B. J. Van der Zwaag, J. C. M. Vernooij, P. R. van Weeren, and W. Back. 2017. "Validation of distal limb mounted inertial measurement unit sensors for stride detection in Warmblood horses at walk and trot." Equine Vet J 49 (4):545-551. doi: 10.1111/evj.12651.

Loscher, D. M., F. Meyer, K. Kracht, and J. A. Nyakatura. 2016. "Timing of head movements is consistent with energy minimization in walking ungulates." Proc Biol Sci 283 (1843). doi: 10.1098/rspb.2016.1908.

van Weeren, P. R., A. J. van den Bogert, and A. Barneveld. 1990. "Quantification of skin displacement in the proximal parts of the limbs of the walking horse." Equine Vet J Suppl (9):110-8. doi: 10.1111/j.2042-3306.1990.tb04746.x.

---

## [Decision Letter · Decision Letter 1]

26 May 2021

PONE-D-20-38036R1

Kinematic gait characteristics of straight line walk in clinically sound dairy cows

PLOS ONE

Dear Dr. Tijssen,

Thank you for submitting your manuscript to PLOS ONE. After careful consideration, we feel that it has merit but does not fully meet PLOS ONE’s publication criteria as it currently stands. Therefore, we invite you to submit a revised version of the manuscript that addresses the points raised during the review process.

Please address the comments of both reviewers through these minor revisions. I concur with the reviewers that some of the aspects explained in the rebuttal letter had not been translated into the body of the revised manuscript.

We look forward to receiving your revised manuscript.

Kind regards,

Angel Abuelo, DVM, MRes, MSc, PhD, DABVP (Dairy), DECBHM

Academic Editor

PLOS ONE

Journal Requirements:

Reviewers' comments:

Reviewer's Responses to Questions

**Comments to the Author**

1. If the authors have adequately addressed your comments raised in a previous round of review and you feel that this manuscript is now acceptable for publication, you may indicate that here to bypass the “Comments to the Author” section, enter your conflict of interest statement in the “Confidential to Editor” section, and submit your "Accept" recommendation.

Reviewer #1: All comments have been addressed

Reviewer #2: All comments have been addressed

2. Is the manuscript technically sound, and do the data support the conclusions?

Reviewer #1: Partly

Reviewer #2: Yes

3. Has the statistical analysis been performed appropriately and rigorously? 

Reviewer #1: N/A

Reviewer #2: I Don't Know

4. Have the authors made all data underlying the findings in their manuscript fully available?

Reviewer #1: Yes

Reviewer #2: Yes

5. Is the manuscript presented in an intelligible fashion and written in standard English?

Reviewer #1: Yes

Reviewer #2: Yes

6. Review Comments to the Author

Reviewer #1: Dear authors,

First my apologies for my late review, some circumstances made me have to go off grid for a few weeks. Nevertheless, below follows my (few) remaining comments.

Second, thank you for addressing my previous comments. You did a great job providing me with detailed answers to my questions and concerns, which clarified for me most things. And I appreciate that “biomechanics” were toned down to “kinematic”, which is more to the point concerning the IMU’s.

However, some of the clarifications from the rebuttal letter were not included in your manuscript. And this (type of discussion) is exactly what I think is lacking and potentially lifts up the manuscript’s value for the readership of the journal. Not to forget about the fact that many people from the dairy domain are unfamiliar with the horse kinematic studies using IMU technology or kinematics in general.

Three points:

1) The change from “limb angle” to “distal limb angle” is a step in the right direction, however I think you could provide a good kinematic definition of what is exactly meant by this in the M&M section. Hence explain the relevance of the specific body segment’s angle and orientation (metatarsus and metacarpus) in the step cycle of the particular limb AND why it is useful to obtain this information. It could be added to the discussion from the perspective of the “clinical gait evaluation” in which multiple IMU could be used, opposed to the “farm setting” where a single IMU is mounted to one of the legs lameness screening, as you state to be the best location. However, single sided IMU’s make it more difficult to show asymmetry, hence a position in the neck might show the best trade-off, or not?

2) Whether it is the clinical or the farm setting, it cannot be neglected in the discussion that machine vision technologies are rapidly evolving to a situation which is easily compared with your technology. And, it is my personal opinion we need your type of studies to be able to explain/verify these new approaches, and vice versa. In addition, it is strange to me to read this opinion in the last line of your manuscript, without it being properly discussed.

3) If no more interpretation/aggregation of data is provided, basically it is left to the reader, then in general there are too many (vertical) displacement figures in my opinion.

a. One figure of all cows with one of the IMU’s is sufficient to show interindividual variations that is similar for all sensors.

b. One cow’s all IMU’s to show the paths of various locations is enough to show the difference between locations.

Minor:

Line 75 kinetic => kinematic

Other than that, no further comments.

Reviewer #2: Thank you for this. The authors adressed all my comments.

see my previous comment (72-73: how the authors can explain the importance of applying the sensors in upper body in comparison to the "traditional" limb positions?)

I suggest to add your your explanation to the text accordingly.

7. PLOS authors have the option to publish the peer review history of their article (what does this mean?). If published, this will include your full peer review and any attached files.

Reviewer #1: **Yes: **P.P.J. van der Tol

Reviewer #2: No

---

## [Author Response · Author response to Decision Letter 1]

1 Jun 2021

Journal Requirements:

We are not aware of referring to any retracted papers.

Reviewers' comments:

Dear reviewers, we would like to thank you for going through the manuscript again. It really improved the manuscript and we wrote down the revisions made accordingly.

Reviewer #1: Dear authors,

First my apologies for my late review, some circumstances made me have to go off grid for a few weeks. Nevertheless, below follows my (few) remaining comments.

Second, thank you for addressing my previous comments. You did a great job providing me with detailed answers to my questions and concerns, which clarified for me most things. And I appreciate that “biomechanics” were toned down to “kinematic”, which is more to the point concerning the IMU’s.

However, some of the clarifications from the rebuttal letter were not included in your manuscript. And this (type of discussion) is exactly what I think is lacking and potentially lifts up the manuscript’s value for the readership of the journal. Not to forget about the fact that many people from the dairy domain are unfamiliar with the horse kinematic studies using IMU technology or kinematics in general.

We checked for discussion points lacking in the manuscript, but we feel all points are currently present. We feel that the kinematic IMU information is indeed valuable information in the dairy domain.

Three points:

1) The change from “limb angle” to “distal limb angle” is a step in the right direction, however I think you could provide a good kinematic definition of what is exactly meant by this in the M&M section. Hence explain the relevance of the specific body segment’s angle and orientation (metatarsus and metacarpus) in the step cycle of the particular limb AND why it is useful to obtain this information. It could be added to the discussion from the perspective of the “clinical gait evaluation” in which multiple IMU could be used, opposed to the “farm setting” where a single IMU is mounted to one of the legs lameness screening, as you state to be the best location. However, single sided IMU’s make it more difficult to show asymmetry, hence a position in the neck might show the best trade-off, or not?

The sagittal limb angle, i.e. protraction – retraction angle, is the most prominent movement of the limbs with the highest range of motion. It is useful to measure this limb angle because they are inertly related to the mechanism the animal uses to move (reverse pendulum model of the walk) and can indicate if the left and right limbs (as well as front and hind) are being used differently. From this angle, we can calculate temporal stride parameters such as stance, swing and all inter-limb related parameters. We have also demonstrated that these angles can be affected by lameness in horses at walk (Serra Braganca et al, 2020). How protraction/retraction are measured and defined varies significantly in the literature. We have chosen this angle as a practical approach because it’s a feasible manner to instrument the limb with a sensor on the metatarsal/-carpus and no substantial skin displacement occurs on this locations. Since, the angles we investigated (max and min) happen at the end of the swing and stance. Both of these events are closely related to the actual entire limb protraction/retraction angle which can be best measured closest to the claw. Using an IMU approach, there is no other solution to access these angles and if not using our approach, the entire limb kinematic could not be accessed.

We were was not aware of stating that the a single sided limb mounted IMU is the best location in regards to leg lameness screening. In our next paper, we will discuss this more deliberately and compare lameness data measured at all IMU locations before concluding which location is best to measure lameness.

2) Whether it is the clinical or the farm setting, it cannot be neglected in the discussion that machine vision technologies are rapidly evolving to a situation which is easily compared with your technology. And, it is my personal opinion we need your type of studies to be able to explain/verify these new approaches, and vice versa. In addition, it is strange to me to read this opinion in the last line of your manuscript, without it being properly discussed.

We agree that it is better to mention this also in the introduction of the manuscript. We have therefore added the sentence “This knowledge is paramount for the development of an automatic recognition method, which can also be used as an early warning system.” to line 72-73.

3) If no more interpretation/aggregation of data is provided, basically it is left to the reader, then in general there are too many (vertical) displacement figures in my opinion.

a. One figure of all cows with one of the IMU’s is sufficient to show interindividual variations that is similar for all sensors.

We agree that we show a lot of figures in the main text. We have therefore moved Fig 9 and 10, showing the vertical displacement of the withers and the back, to the supplementary materials. We have left the description in the manuscript as it is and only refer to the figures in the supplementary materials. Since the left and right tuber coxae show a different type of pattern compared to the other upper body locations, we would like to leave this figure in the main document. We also have moved Fig 12 and Fig 13, showing the vertical displacement of the head and collar, to the supplementary materials and only describe that these locations show a less consistent pattern.

b. One cow’s all IMU’s to show the paths of various locations is enough to show the difference between locations.

This is depicted in Fig 14.

Minor:

Line 75 kinetic => kinematic

Other than that, no further comments.

Thank you for noticing

Reviewer #2: Thank you for this. The authors addressed all my comments.

see my previous comment (72-73: how the authors can explain the importance of applying the sensors in upper body in comparison to the "traditional" limb positions?)

I suggest to add your explanation to the text accordingly.

We have added this explanation to the introduction in line 71-77: “Given the visibility of these body landmarks, even for a cow among herd mates, it would be of future interest to explore if asymmetries in the vertical excursions of the upper body could prove sensitive for detection of lame animals. This can be of particular interest in the development on automated lameness detection systems based on computer vision, where challenging sensor attachment is superfluous, but optical occlusion of limbs is a challenge. Prerequisite for such studies is the knowledge of normal walking gait.”

---

## [Editor Report · Decision Letter 2]

7 Jun 2021

Kinematic gait characteristics of straight line walk in clinically sound dairy cows

PONE-D-20-38036R2

Dear Dr. Tijssen,

We’re pleased to inform you that your manuscript has been judged scientifically suitable for publication and will be formally accepted for publication once it meets all outstanding technical requirements.

Kind regards,

Angel Abuelo, DVM, MRes, MSc, PhD, DABVP (Dairy), DECBHM

Academic Editor

PLOS ONE
---

## [Editor Report · Acceptance letter]

24 Jun 2021

PONE-D-20-38036R2 

Kinematic gait characteristics of straight line walk in clinically sound dairy cows 

Dear Dr. Tijssen:

I'm pleased to inform you that your manuscript has been deemed suitable for publication in PLOS ONE. Congratulations! Your manuscript is now with our production department. 

Kind regards, 

on behalf of

Dr. Angel Abuelo 

Academic Editor

PLOS ONE